# SkillFactory: Self-Distillation for Learning Cognitive Behaviors

**Zayne Sprague**♠, **Jack Lu**♠, **Manya Wadhwa**♠, **Sedrick Keh**◇,
**Mengye Ren**♠, **Greg Durrett**♠

♠New York University, ◇Toyota Research Institute
zrs2020@nyu.edu

## Abstract

Reasoning models leveraging long chains of thought employ various cognitive skills, such as verification of their answers, backtracking, retrying by an alternate method, and more. Previous work has shown that when a base language model exhibits these skills, training that model further with reinforcement learning (RL) can learn to leverage them. How can we get models to leverage skills that aren't exhibited by base models? Our work, SkillFactory, is a method for fine-tuning models to roughly learn these skills during a supervised fine-tuning (SFT) stage prior to RL. Our approach does not rely on distillation from a stronger model, but instead uses samples from the model itself, rearranged to provide training data in the format of those skills. These "silver" SFT traces may be imperfect, but are nevertheless effective for priming a model to acquire skills during RL. Our evaluation shows that (1) starting from SkillFactory SFT initialization helps a model to generalize to harder variants of a task post-RL, despite lower performance pre-RL; (2) cognitive skills are indeed used by the model; (3) RLed SkillFactory models are more robust to regression on out-of-domain tasks than RLed base models. Our work suggests that inductive biases learned prior to RL help models learn robust cognitive skill use[1].

## 1 Introduction

Modern large language models (LLMs) increasingly demonstrate the ability to acquire and apply a variety of cognitive behaviors we can call "skills." These include capabilities such as systematically exploring a solution space, verifying outputs, and retrying with alternative strategies (Marjanović et al., 2025). Such skills are particularly valuable for reasoning, as they enable models to explore different paths to a solution rather than relying on a single attempt (Bogdan et al., 2025). Indeed, many of the major gains in reasoning-focused LLMs in the recent literature can be traced to better elicitation of these skills during inference time, demonstrating that skill acquisition itself has become a primary driver of progress in reasoning (Jaech et al., 2024; Guo et al., 2025; Abdin et al., 2025).

Reinforcement learning (RL) has proven to be a powerful paradigm for unlocking many of these capabilities (Guo et al., 2025). If a model already demonstrates these skills, or is equipped with them through distillation or continued pre-training, then RL can further reinforce these behaviors (Gandhi et al., 2025). However, these approaches require access to superior models (Muennighoff et al., 2025; Guha et al., 2025), significant training (Yeo et al., 2025), custom pre-training data, or a complex mix of all of these. Additionally, these methods are often evaluated according to how much they improve models' downstream evaluation results after SFT; it is unclear whether such improvements reflect a better ability to learn skills during RL.

In this work, we propose **SkillFactory**, a framework to instill these behaviors into models and unlock large gains from RL *without* distilling from a larger model. Through prompting and restructuring of the samples into a structured output, we can construct "silver" traces that demonstrate a model verifying its outputs and retrying based on failures. See Figure 1 for an example of how correct and

---

[1] All code and data can be found at https://github.com/Zayne-sprague/SkillFactory

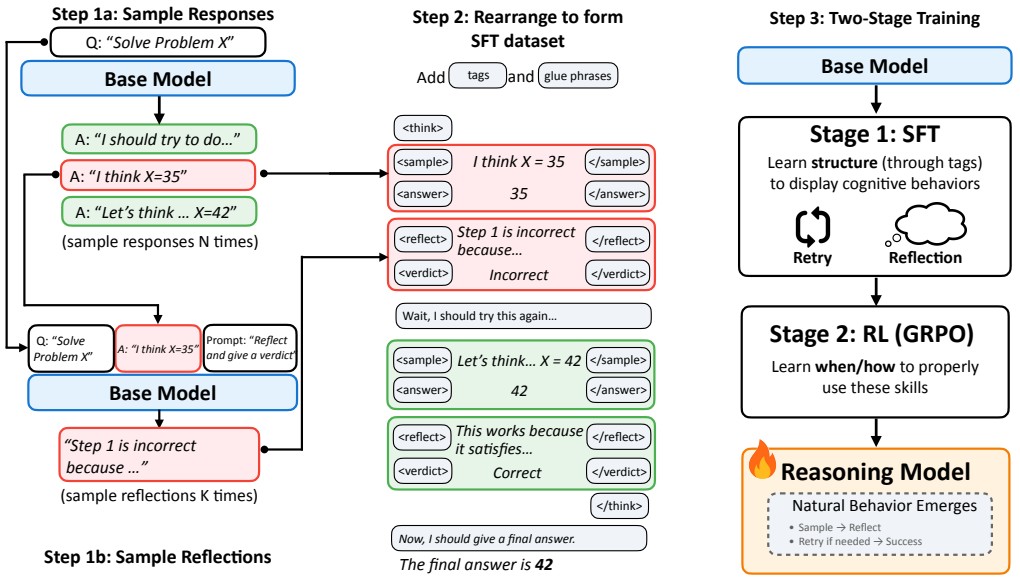

Figure 1: SkillFactory framework. We obtain responses and reflection traces using a model's own sampled reasoning, then rearrange them to demonstrate reasoning skills. A model SFTed on this data is an effective starting point for RL, yielding better performance and more skill usage post-RL.

incorrect attempts by a model to solve the problem can be remixed into a trace exhibiting verification. A model trained on this data with supervised fine-tuning (SFT) is not yet calibrated to use these skills effectively; however, past work suggests that focusing on the structure alone of a skill can be highly effective (Li et al., 2025), and the model may be primed for effective RL. The RL stage hones the skills instilled into the model, improving both how they are used and where. Crucially, higher performance prior to RL does not necessarily imply higher performance post-RL; priming to use the appropriate skills may be more important than having maximally learned the task.

**Contributions** We demonstrate that (1) across two training settings (Countdown and OpenThoughts (Guha et al., 2025)), models can acquire complex reasoning skills from their own rearranged outputs without requiring stronger teacher models; (2) SkillFactory initialization enables generalization to harder task variants and novel domains post-RL, matching or exceeding the performance of strong baselines; and (3) SkillFactory models show greater resilience to catastrophic forgetting and regression of performance on out-of-domain tasks.

## 2 BACKGROUND AND MOTIVATION

### 2.1 COGNITIVE SKILLS IN LLMS

LLMs take in an input $\mathbf{x}$ and place a distribution $p(\mathbf{y} \mid \mathbf{x})$. For the tasks we consider, we assume a final answer can be extracted via a process $a = \texttt{extract}(\mathbf{y})$ (e.g., if it is embedded in <answer> tags). Large *reasoning* models fit in this framework but are characterized by two differences: (1) they exhibit the use of reasoning skills rather than simple "linear" solving processes; (2) as a result, their outputs $\mathbf{y}$ are typically much longer. Past work describes a number of cognitive skills useful for reasoning (Gandhi et al., 2025). In this work, we focus on the following two:

1. **Retrying:** A prefix $\mathbf{y}_{<i}$, where $i$ is the length in tokens, ends in an answer $\tilde{a} = \texttt{extract}(\mathbf{y}_{<i})$. The model decides to restart its inference, generating tokens like "*Wait, let me rethink this...*" and generating completion $\mathbf{y}_{\geq i}$ with potentially little connection to what came before.

2. **Reflection:** A prefix $\mathbf{y}_{<i}$ ends in an answer $\tilde{a} = \texttt{extract}(\mathbf{y}_{<i})$. The model enters a separate process of verifying $\tilde{a}$, generating tokens $\mathbf{y}_{v(\tilde{a})}$ focused on evaluating the answer.

Together, these methods guide the model to generate long chain-of-thoughts beyond a single attempt, leading to more robust reasoning.

**Existing usage of skills and the need for SkillFactory** A central finding of Gandhi et al. (2025) is that some base language models already exhibit these skills in some form. Figure 2 shows an example of this for the Countdown number puzzle, where the task is to combine a set of input numbers using the four basic arithmetic operations ($+, -, \times, \div$) to reach a target number. A red highlight shows the model verifying the outcome of the computation as incorrect; highlighted in blue is an instance where the model restarts and tries to find another solution.

Two fundamental observations underlie our work. First, **these skills surface less consistently when incidentally expressed in natural language**. We will see in our results that SkillFactory consistently leads to longer traces exhibiting phenomena like verification and retries to a higher extent than the base model, particularly on out-of-domain tasks.

Second, **there are advantages to explicitly marking skills' usage**. Our format, shown in Figure 3, has reflection *explicitly demarcated* by a tag. The reflection process also reruns the computation, potentially mitigating errors that happened during the initial search. Retrying is similarly marked by a tag.

Finally, SkillFactory allows us to impart reasoning skills that are unattested in the base model. We therefore view this work as providing a platform for shaping cognitive behaviors of LLMs across a variety of tasks. Correcting the use of cognitive behaviors can be a pathway towards getting models to avoid overthinking (excessive verbosity) (Sui et al., 2025) and underthinking (prematurely abandoning solution paths) (Wang et al., 2025). While some argue for fundamental limitations in reasoning models due to problem complexity (Shojaee et al., 2025), we argue that skills offer a way to explore solution paths and exit them effectively when learned correctly.

[…]
Let's try a different combination:

1. First, add 98 and 20:
   Step 1: 98 + 20 = 118
2. Then, add 44 to the result:
   Step 2: 118 + 44 = 162

*This is also not equal to 102. Let's try a simpler approach with a different order:*

*1. First, subtract 20 from 98:*
   *Step 1: 98 - 20 = 78*
*2. Then, add 44 to the result:*
   *Step 2: 78 + 44 = 122*
[…]

Figure 2: Trace from Countdown exhibiting implicit reflection and retrying.

**Existing Approaches to Eliciting Reasoning Skills** Current methods for developing reasoning capabilities in language models can be broadly categorized into three main approaches. First, simply doing RL with sparse rewards can surface reasoning behaviors latent in the base model (Shao et al., 2024; Yu et al., 2025; Liu et al., 2025). This approach relies heavily on a strong base model, and these skills may fail to emerge naturally when not sufficiently represented in the pre-training data; our results show that pure RL does not yield robust skill use in cross-task generalization. Second, distillation from stronger models (Muennighoff et al., 2025; Ye et al., 2025; Guha et al., 2025) enables SFT on traces showing advanced reasoning, though past approaches assume access to superior models and often struggle to generalize beyond the domains of the distilled data (Gudibande et al., 2024; Kalai et al., 2025). Third, targeted data curation, through continual pre-training on backtracking examples (Gandhi et al., 2025), hand-crafted reasoning chains for in-context learning (Pang et al., 2025), or Monte Carlo tree search rollouts (Kim et al., 2025, ASTRO), have shown promise in instilling specific cognitive skills before or during fine-tuning. SkillFactory is similar to these methods, but focuses on generating data entirely from the base model and highlights that structure is key for the generalization of consistent skill use.

## 2.2 TASKS: PLANNING, SEARCH, AND COMPUTATION

The usefulness of cognitive skills varies across tasks. While a skill like verification can in principle be used anywhere, it is more effective on "NP-complete"-like tasks: those that are easier to check than to generate answers for. We call this category of tasks **search-focused** tasks, which are a subset of tasks we evaluate on in this work. A full set of tasks can be found in Section 4.2.

Search-focused tasks are those like Countdown (Figure 2). The space of possible responses is usually large, and an LLM is expected to execute search in its context to find an answer. Verification and

retrying are *naturally exhibited* by models, although not in all traces, and verification is highly effective, since the solutions are easier to check than they are to find. When models are trained on search-focused tasks that naturally elicit skills like verification and retry, we find a tradeoff: light training fails to transfer these skills beyond similar search tasks, while heavier training improves those skills but degrades performance on broader, out-of-distribution tasks.

Other tasks such as multiplication and CommonsenseQA (Talmor et al., 2019) may predominantly require skills other than search, such as forward-chaining of mathematical operations (GSM8K). LLMs at the scale we experiment on are still prone to making mistakes in these tasks. In spite of this, verification and retrying are *not naturally exhibited* despite potentially being beneficial.

# 3 SKILLFACTORY

SkillFactory has three pieces, depicted in Figure 1. (1) Data curation: uses inference on a base model in combination with heuristics tied to each cognitive skill of interest. (2) Supervised fine-tuning on these traces. Unlike other distillation approaches, we don't expect performance to increase in this step; we are only trying to achieve a better starting point for RL. (3) Reinforcement learning: We use off-the-shelf RL algorithms such as GRPO (Shao et al., 2024; Marjanović et al., 2025), combined with sparse rewards based on correctness. We focus on the data curation stage in this section.

We generate SkillFactory data in three steps: sampling diverse solutions from the base model, generating reflections that assess those solutions, and combining them into structured traces that exhibit explicit retry and verification behaviors. Throughout this process, we use $\mathbf{y}$ to denote solution attempts and $\mathbf{r}$ to denote reflections. Algorithm 1 provides a detailed algorithm outlining the data curation process for SkillFactory.

---

**Algorithm 1 SkillFactory Trace Construction**. All values of the parameters used in the Trace Construction algorithm can be found in Table 12 of the Appendix.

---

**Require:** Dataset $D_T = \{(\mathbf{q}_i, \mathbf{a}_i)\}$, base model $\mathcal{M}$, prompts $P_{\text{solve}}, P_{\text{reflect}}$
**Ensure:** Training set $\mathcal{D}_{\text{SFT}}$
1: $\mathcal{D}_{\text{SFT}} \leftarrow \emptyset$
2: **for** each question $(\mathbf{q}_i, \mathbf{a}_i) \in D_T$ **do**
3:     // Generate solution-reflection pairs
4:     Sample solutions: $\mathcal{Y} \leftarrow \{\mathbf{y}_j \sim \mathcal{M}(\mathbf{q}_i \mid \mathbf{p}) : \mathbf{p} \in P_{\text{solve}}, j \in \{1, 2, \ldots, N_{\text{sample}}\}$
5:     Generate reflections: $\mathcal{R} \leftarrow \{\mathbf{r} \sim \mathcal{M}(\mathbf{q}_i, \mathbf{y} \mid P_{\text{reflect}}) : \mathbf{y} \in \mathcal{Y}, \text{verdict}(\mathbf{r}) = \texttt{correct}(\mathbf{y}, \mathbf{a}_i)\}$
6:     $\mathcal{Y}^+ \leftarrow \{(\mathbf{y}, \mathbf{r}) : \texttt{correct}(\mathbf{y}, \mathbf{a}_i) = \text{True}\}$         ▷ correct pairs
7:     $\mathcal{Y}^- \leftarrow \{(\mathbf{y}, \mathbf{r}) : \texttt{correct}(\mathbf{y}, \mathbf{a}_i) \neq \text{True}\}$         ▷ incorrect pairs
8:     **while** $|\mathcal{Y}^+| > 0$ **do**
9:         // Determine trace length
10:         $n^+ \leftarrow \min(\text{Uniform}([1, L_{\max}]), |\mathcal{Y}^+|)$
11:         $n^- \leftarrow \min(\text{Uniform}([0, n^+ - 1]), |\mathcal{Y}^-|)$
12:         // Sample solution-reflection pairs
13:         $T^+ \leftarrow$ sample $n^+$ items from $\mathcal{Y}^+$ without replacement
14:         $T^- \leftarrow$ sample $n^-$ items from $\mathcal{Y}^-$ without replacement
15:         // Build trace, ensuring that it ends on a correct solution
16:         $\mathbf{trace} \leftarrow \text{shuffle}(T^- \cup T^+[1 : n^+ - 1]) \cup \{T^+[n^+]\}$     ▷ Append last correct
17:         // Format into training instance
18:         $\mathcal{D}_{\text{SFT}} \leftarrow \mathcal{D}_{\text{SFT}} \cup \{\texttt{format}(\mathbf{q}_i, \mathbf{trace})\}$
    **return** $\mathcal{D}_{\text{SFT}}$

---

**Solution Generation** First, for each question $\mathbf{q}_i$ in our task dataset $D_T = \{(\mathbf{q}_i, \mathbf{a}_i)\}_{i=1}^n$, we sample $N_{\text{sample}}$ solution attempts from our base model $\mathcal{M}$. To encourage diversity, we use a set of four different chain-of-thought prompts $P_{\text{solve}}$. For each prompt, we sample 16 responses, yielding a solution set $\mathcal{Y}$ of 64 attempts per question. The full set of prompts can be found in Appendix D.2.

Each solution $\mathbf{y} \in \mathcal{Y}$ is automatically verified: we use $\texttt{extract}(\mathbf{y})$ to parse the final answer from the solution and check if it matches the ground truth $\mathbf{a}_i$. Since SkillFactory prompts the model to enclose its final answer in `<answer>` tags, our $\texttt{extract}()$ function leverages these tags for parsing.

We define $\mathtt{correct}(\mathbf{y}, \mathbf{a}_i) = \mathbb{1}[\mathtt{extract}(\mathbf{y}) = \mathbf{a}_i]$ to indicate whether a solution is correct. This gives us a pool of both correct and incorrect solutions; both are needed to teach the model self-correction.

**Reflection Generation**   Next, we prompt $\mathcal{M}$ to reflect on each solution attempt using a reflection prompt $p_{\text{reflect}}$. A reflection $\mathbf{r}$ critiques the reasoning in solution $\mathbf{y}$ and predicts its correctness, $\mathtt{correct}(\mathbf{y}, a_i)$. We use $\mathtt{verdict}(\mathbf{r})$ to extract this prediction from the reflection text. Just like with the answer tags, SkillFactory also prompts the model to use <verdict>...</verdict> tags when generating reflections, which we then use for parsing the verdicts. A valid reflection is one where $\mathtt{verdict}(\mathbf{r}) = \mathtt{correct}(\mathbf{y}, \mathbf{a}_i)$. The reflection prompts can be found in Appendix D.3.

```
User: [question]
Assistant: <think>
[Attempt 1]
Reflect: "Wrong because..."
Let me try again.
[Attempt 2]
Reflect: "Need to verify..."
...
[Final correct attempt]
[Reflection: "This looks correct..."]
</think>
Answer: [final answer]
```

Figure 3: SkillFactory training trace with self-reflection and retry.

We sample four reflections per solution but keep only those where $\mathtt{verdict}(\mathbf{r}) = \mathtt{correct}(\mathbf{y}, \mathbf{a}_i)$, reflections that accurately judge whether the solution succeeded or failed. The result is a set $\mathcal{R}$ of valid reflections paired with their corresponding solutions.

**Trace Construction**   Finally, we assemble solution-reflection pairs into training traces. We partition our pairs into correct ($\mathcal{Y}^+$) and incorrect ($\mathcal{Y}^-$). For each trace, we:

- Sample $n^+$ correct pairs and $n^-$ incorrect pairs
- Shuffle all but one correct pair to create a mixed sequence
- Append the remaining correct pair to ensure success at the end
- Format the sequence using $\mathtt{format}()$, which wraps each solution-reflection pair in tags and adds transition phrases; see Figure 3.

This creates traces where the model attempts a problem, reflects on its work, tries again if needed, and always eventually succeeds. The $\mathtt{format}()$ function applies the template shown in Figure 3, interleaving solutions with reflections in <sample> and <reflect> tags respectively. Pairs of samples and their reflections are concatenated together with phrases like "*Let me reconsider*". By training on these restructured outputs, we prime the model to employ these skills during RL. A full list of phrases used to stitch together the pairs can be found in Appendix D.1.

## 4   EXPERIMENTAL SETUP

We evaluate SkillFactory in two main settings. First, we train models on Countdown and evaluate on a suite of reasoning tasks. Second, we train models on the OpenThoughts dataset and evaluate on challenging math and science datasets. Our experiments use three different base models: Qwen2.5-1.5B-Instruct (Team, 2024), Qwen2.5-7B-Instruct (Team, 2024), and Olmo-3-7B-Instruct (Olmo Team, 2025).

### 4.1   BASELINES

We evaluate SkillFactory against four baselines, each representing a different paradigm for developing reasoning models as outlined in Section 2. Most baselines can be thought of as "warm-starting" the policy model, imparting some key knowledge that is hoped to be enhanced during RL, thereby avoiding the "cold-start" problem (Gandhi et al., 2025; Guo et al., 2025).

**RL Only**   We directly train the base model using only reinforcement learning with binary correctness rewards. We use the same GRPO setup as SkillFactory, but start from the base model.

**BOLT (external data curation)**   Similar to BOLT (Pang et al., 2025), we (1) Sample 10 in-context learning examples from a strong reasoning model (Claude Sonnet 4), (2) prompt an LLM (GPT-4o-mini) with ICL to generate reasoning traces for new problems, creating synthetic SFT data,

and (3) train the resulting model using GRPO. We provide additional details in Appendix F. Our implementation uses different models than BOLT for data creation and uses GRPO instead of DPO.

**Distillation (learning from strong models)**   We also evaluate distillation (Muennighoff et al., 2025; Ye et al., 2025; Guha et al., 2025), where we train on traces from a more capable model. We prompt R1 to solve problems from our training set and collect its generated reasoning traces. We perform SFT on these traces. In **R1 Distill** → **GRPO**, we then further fine-tune with RL. Because this method relies on the existence of a stronger model, we treat it separately from other baselines.

**STaR (learning from correct outputs)**   Finally, we compare with STaR (Zelikman et al., 2022), another self-distillation method. STaR iteratively samples from the base model, checks if the answer is correct, and subsequently uses it to train the model if the answer is correct. We perform this for our base model then train with RL.

## 4.2   TASK SETUP AND EVALUATION

**Countdown** requires the model to take a set of input numbers and apply mathematical operations $+, -, \times, \div$ to reach a target. The inputs can be used in any order, but each number can be used at most once. The $N$ arg variant of this task has $N$ numbers to combine. We also explore a variant of this task called **Letter Countdown (CD)**, which requires the model to assemble scrambled letters into a word of a specified length. For example, the model may be given "ppale" as input, and the model must create a valid English word using only those letters and must be of length 5 characters such as "apple". Correctness in this task is gauged by the length of the unscrambled word submitted by the model, that only the given letters were used, and that the word exists in an English dictionary. We guarantee that an answer exists. We consider both $N = 4$ and $N = 5$.

**Acryonym Generation** tasks the model with taking as input a list of words, where the model must take the first letter from a subset of words and put those letters together to create a valid english word of size $N$. For example, the model may be given "Air Ball People Places Deck Left True Never Eat" where the model needs to extract a correct subset of words and their first letters "a p p l e" and then recognize the valid word "apple". We consider $N = 4, 5$ in this work. We ensure that every set of words yields at least one valid acronym that could be created from them.

**Multiplication** requires the model to multiply two numbers of $N$ digits each and return the answer. In this work we consider 2, 3, 4, and 5 digit multiplication tasks. Previous work showed this task to be hard for LLMs (Dziri et al., 2023).

We also evaluate on **CommonsenseQA (CSQA)** (Talmor et al., 2019), a multiple choice dataset, and **GSM8K** (Cobbe et al., 2021), a dataset of grade-school math problems.

For the models trained on OpenThoughts data, we evaluate on more challenging math and science datasets including **GPQA** (Rein et al., 2024), **AIME 2025** (MAA, 2025), **AMC** (MAA, 2023), and **Math500** (Lightman et al., 2023).

All tasks we evaluate on, with the exception of CSQA, GSM8K, and the harder math datasets, have multiple difficulty levels, or ways for us to test generalization from easier tasks to harder variants of the same task (such as increasing the amount of input numbers to Countdown). We treat CSQA and GSM8k as generalization to out-of-domain tasks that are less related to the other tasks to help capture any regressions in the capabilities of the model and see how well these methods generalize. Details on our decoding parameters and sample rates for each dataset can be found in Appendix A.3.

## 4.3   TRAINING SETTINGS

We test SkillFactory in two different training regimes. The first focuses on Countdown-3arg and is the focus of our primary experiments. In this setting we use 4,000 rows of Countdown-3arg for creating SFT data. We then train using RL on an additional held-out set of 1,000 Countdown-3arg questions. This simulates targeted training on a very specific and narrow domain in which it would be easy for the model to overfit. We fine-tune Qwen2.5-1.5B-Instruct (Team, 2024), Qwen2.5-7B-Instruct (Team, 2024), and Olmo-3-7B-Instruct (Olmo Team, 2025) for these experiments.

Second, we explore training on a subset of the **OpenThoughts** dataset (Guha et al., 2025), a dataset of questions and traces from QwQ (Team, 2024). We experiment with using 1,000 and 10,000 rows

Table 1: Performance on Countdown and OOD tasks for Qwen2.5-1.5B-Instruct models trained on Countdown-3arg. Evaluations here are average across held-out difficulties: Countdown (4,5,6-arg), Acronym (4,5), Letter CD (4,5), Long Multiplication (2,3,4,5-digit). Highlighted columns use larger models for the SFT data.

| Model | Countdown | Acronym | Letter CD | Mult | CSQA | GSM8k | Overall |
|---|---|---|---|---|---|---|---|
| Qwen2.5 1.5B Instruct | 1.9 | 6.9 | 10.4 | 29.8 | 55.7 | 59.2 | 27.3 |
| BOLT | 0.5 | 6.2 | 5.5 | 15.1 | 46.7 | 23.4 | 16.2 |
| R1 Distill | 11.7 | 9.4 | 8.8 | 32.4 | 56.6 | 62.9 | 30.3 |
| STaR | 2.6 | 4.0 | 7.3 | 22.1 | 55.4 | 31.1 | 20.4 |
| SkillFactory | 2.8 | 3.0 | 8.7 | 32.4 | 47.1 | 59.1 | 25.5 |
| RL-Only | 15.8 | 8.7 | 12.5 | 24.4 | 62.6 | 67.7 | 31.9 |
| BOLT → GRPO | 13.7 | **12.3** | 13.1 | 26.6 | 62.8 | 69.7 | 33.0 |
| R1 Distill → GRPO | 21.2 | 6.0 | **14.4** | **37.1** | **63.8** | **72.9** | **35.9** |
| STaR → GRPO | 9.7 | 9.8 | 9.2 | 23.2 | 60.5 | 68.6 | 30.2 |
| SkillFactory → GRPO | **25.1** | 12.1 | 12.8 | 35.0 | 60.8 | 68.2 | 35.7 |

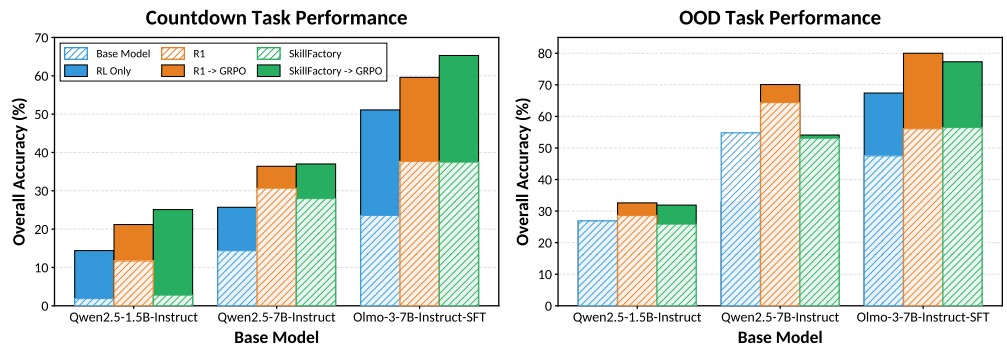

Figure 4: Results showing performance of different models trained using SkillFactory. Left: Averaged overall accuracy on the harder variants of Countdown-(4, 5, 6arg) for models trained on Countdown-3arg only. Right: Averaged overall accuracy of the held-out tasks (Acronym, Letter CD, Multiplication, CSQA, GSM8k) for models trained on Countdown-3arg only.

from the dataset for creating SFT data. For SkillFactory we follow the same procedure outlined in Section 3, with an additional modification that we include a new set of prompts that hint at the right answer to help the model solve challenging questions. We then RL the models using an additional 10,000 held-out rows from OpenThoughts. We compare SkillFactory with distillation from QwQ with GRPO along with using GRPO only (RL only). We fine-tune one model, Qwen2.5-7B-Instruct (Team, 2024), for this experiment. We train with a max context length of 4,096 and evaluate at 16,384. Full hyperparameters for both experiments are provided in Appendix A.1. Details on OpenThoughts, including how we extract data and sample, can be found in Sections D.4 and D.5 of the Appendix.

## 5 RESULTS

We separate our results into three evaluations designed to stress generalization, robustness, and capability gains. First, we study **easy-to-hard generalization** on the Countdown family: models are trained only on COUNTDOWN-3ARG for both SFT and RL and evaluated on held-out harder variants (4–6 arguments). Second, we evaluate **out-of-domain (OOD) generalization** on tasks never seen during training, such as Letter Countdown, Acronym, Multiplication, CSQA, and GSM8K. These results are summarized in Table 1 and Figure 4. Finally, for our models in the OpenThoughts setting, we measure **reasoning capability on challenging math benchmarks** (GPQA, AIME25, AMC, Math500) (Table 2). Across all settings, we compare SkillFactory with strong baselines including RL-only, STaR, BoLT, and R1 Distillation, with all baselines having an SFT and RL stage. Further ablations of SkillFactory as well as tables for the raw accuracies of each experiment can be found in sections B and C of the Appendix.

Table 2: Performance of models trained on OpenThoughts data with either 1k or 10k rows of SFT data across challenging math datasets. All models have been trained with SFT and GRPO (RL).

| Model | GPQA | AIME 25 | AMC | Math500 | **Overall** |
|---|---|---|---|---|---|
| RL Only | 53.8 ± 1.6 | 5.4 ± 1.2 | 33.5 ± 0.8 | 59.1 ± 0.8 | 38.0 |
| QwQ with 1k rows | 48.5 ± 1.7 | 10.6 ± 1.4 | 19.9 ± 0.8 | 55.2 ± 0.9 | 33.5 |
| QwQ with 10k rows | **59.5** ± 1.5 | **15.3** ± 1.0 | 36.5 ± 0.9 | 58.6 ± 0.8 | **42.5** |
| SkillFactory with 1k | 56.7 ± 1.5 | 9.7 ± 1.4 | **37.5** ± 0.8 | **64.6** ± 0.7 | 42.1 |
| SkillFactory with 10k rows | 57.9 ± 1.5 | 7.3 ± 1.2 | 35.2 ± 0.7 | 61.9 ± 0.7 | 40.6 |

## 5.1 SKILLFACTORY ENABLES EASY-TO-HARD GENERALIZATION

Table 1 shows that SkillFactory consistently outperforms alternative methods when generalizing from Countdown-3arg to harder variants (4–6 arguments). SkillFactory → GRPO achieves 25.1%, the highest accuracy among all methods, outperforming the next strongest baseline, R1 Distill → GRPO (21.2%), by +3.9 points. In contrast, STaR provides little benefit in this harder regime, performing similarly to the base model before RL and underperforming after RL, whereas SkillFactory improves on RL-only by 9.3%.

Although R1 Distill achieves much higher SFT accuracy than SkillFactory (11.7% vs. 2.8%), this relationship reverses after RL: SkillFactory → GRPO overtakes R1 Distill → GRPO. This suggests that **stronger SFT task solving does not reliably translate into better post-RL performance**. Figure 4 left side confirms this trend for Countdown across three models (Qwen2.5-1.5B, Qwen2.5-7B, OLMo-3-7B). In all cases, SkillFactory outperforms RL-only and matches or exceeds R1 Distill → GRPO.

## 5.2 SKILLFACTORY MAINTAINS ROBUSTNESS OUT-OF-DOMAIN

Table 1 also reports OOD accuracy on tasks never seen during training. R1 Distill → GRPO slightly surpasses SkillFactory → GRPO overall (35.9% vs. 35.7%). However, SkillFactory performs well on average. Figure 4 right side provides additional insight into these OOD trends. We observe that R1 Distill → GRPO often yields strong gains, particularly on larger backbones such as Qwen2.5-7B, likely due to the breadth of latent knowledge and diverse reasoning heuristics encoded in the R1 traces. However, the gap from base models to RLed R1 models is substantially closed in two of three models.

## 5.3 SKILLFACTORY IMPROVES COMPLEX MATHEMATICAL REASONING

We next evaluate whether SkillFactory enhances reasoning capabilities on challenging math datasets. Using Qwen2.5-7B-Instruct, we train on subsets of the OpenThoughts dataset varying the size of the SFT data from 1k to 10k and evaluate on GPQA, AIME25, AMC, and Math500. Table 2) shows that at the 10k scale, SkillFactory reaches an overall score of 40.6%, closely approaching QwQ distillation (42.5%). At the 1k scale, SkillFactory performs competitively across tasks and **surpasses QwQ distillation on AMC (37.5%) and Math500 (64.6%)**, two benchmarks not explicitly targeted in the original OpenThoughts curation. In contrast, QwQ distillation exhibits degradation on Math500 relative to the base model even at 10k.

We note that SkillFactory's performance slightly decreases from 1k to 10k examples (42.1% → 40.6%). We believe additional SFT does not help SkillFactory because the core skills are already learned early, unlike in distillation, where models learn new strategies and knowledge from the teacher.

## 6 BUDGET FORCING

SkillFactory benefits from structured tags (`<sample>`, `<reflect>`) that let the model search, restart, and validate its answers. To test whether it can exploit more "thinking time," we apply a simple budget-forcing intervention (Muennighoff et al., 2025) at inference time. First, the model generates with a 4,096-token budget (matching RL training), then we append a model-specific trigger phrase

Table 3: Performance breakdown on out-of-distribution tasks. "Std" indicates results prior to budget forcing, and "BF" indicates results with budget-forcing for that model.

| Task | RL Only | | | R1 Distill | | | SkillFactory | | |
|---|---|---|---|---|---|---|---|---|---|
| | Std | BF | Δ | Std | BF | Δ | Std | BF | Δ |
| Countdown | 13.8 | 15.0 | 1.2 | 7.1 | 11.8 | 4.7 | 17.5 | 22.8 | **5.3** |
| Acronym | 10.2 | 8.0 | -2.3 | 9.1 | 11.0 | **1.9** | 10.8 | 10.5 | -0.2 |
| CommonsenseQA | 62.8 | 62.8 | 0.1 | 50.9 | 52.1 | **1.3** | 60.9 | 59.8 | -1.0 |
| GSM8k | 68.3 | 68.8 | **0.5** | 51.3 | 49.6 | -1.7 | 67.7 | 66.1 | -1.6 |
| Letter Countdown | 12.0 | 11.9 | -0.2 | 7.0 | 7.8 | **0.8** | 14.6 | 12.1 | -2.5 |
| Multiplication | 24.6 | 31.4 | **6.9** | 25.8 | 25.1 | -0.7 | 35.2 | 36.6 | 1.3 |
| **Overall** | 24.2 | 26.3 | 2.1 | 19.9 | 21.2 | 1.2 | 28.7 | 29.7 | 1.0 |

**Response Length Distributions**

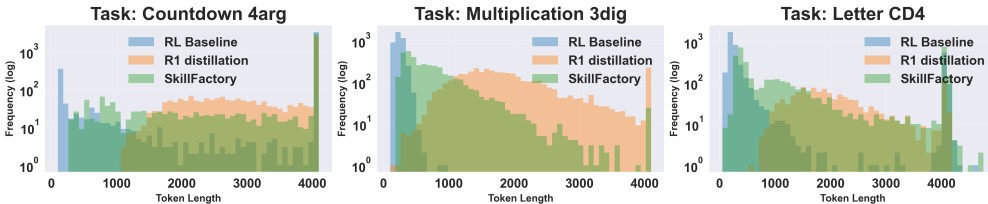

Figure 5: Token length distribution for three tasks for responses given by (a) RL Baseline, (b) R1 distillation, (c) SkillFactory. SkillFactory induces the base model to generate much longer thinking traces, making the distribution of lengths much closer to that of an R1-distilled model.

(for SkillFactory, a `<sample>` tag before the closing `</think>` tag) to request another reasoning attempt and allow continuation up to 8,192 tokens total.

Table 3 reports results when budget forcing is used on the test set. On Countdown, SkillFactory gains +5.3 points (17.5→22.8), outpacing RL-only (+1.2) and R1 distillation (+4.7). RL-only, however, benefits most on multiplication (+6.9, 24.6→31.4) compared to SkillFactory's smaller improvement (+1.3, 35.2→36.6), likely because SkillFactory already performs multiple retries and verifications during standard inference. We observe that improvements come from more effectively using a large output context, which SkillFactory is effective at due to its baked-in cognitive behaviors. We also note that one source of improvement is when a model is producing a degenerate output (looping the same piece of thinking repeatedly), and budget forcing with an explicit tag allows us to break out of this loop.

## 7 ANALYSIS

**Skill Usage** Table 4 shows an analysis of the SkillFactory traces: the average number of explicit answer attempts (final answers given in answer tags), the average number of explicit reflections (explicit reflection and verification done in reflection tags), and the F1 of the verifier steps, broken down by correct class and incorrect class. That is, in Countdown-3arg, we see the SkillFactory verifier achieve an F1 of 0.96 when the answer it proposes is truly correct and an F1 of 0.92 when the answer it proposes is wrong.

Table 4: Number of explicit answer attempts, explicit reflections and the verification F1 for the correct and incorrect classes (represented by (correct/incorrect)) for Skill Factory and the No Sample Order ablation.

| | SkillFactory | | | No Sample Order | | |
|---|---|---|---|---|---|---|
| | #Ans | #Ref | F1 | #Ans | #Ref | F1 |
| CD 3arg | 1.59 | 1.24 | 0.96 / 0.92 | 2.37 | 1.44 | 0.99 / 0.97 |
| CD 4arg | 2.34 | 7.13 | 0.65 / 0.97 | 10.65 | 9.40 | 0.58 / 0.97 |
| Letter CD 4o | 2.11 | 1.78 | 0.34 / 0.82 | 3.01 | 1.81 | 0.22 / 0.65 |
| Mult 3dig | 2.19 | 1.86 | 0.35 / 0.81 | 3.68 | 2.63 | 0.22 / 0.74 |

Reflection is broadly effective: the "incorrect" class F1 values are all above 0.8, meaning that wrong answers are correctly rejected. Reflection generalizes to other domains and scales with task diffi-

culty: Countdown-4arg exhibits more reflection than Countdown-3arg. Cases where performance is lower, such as Letter Countdown, usually reflect weaknesses of the model itself; for instance, the model exhibits uncertainty about what is and isn't an English word, suggesting a limitation of our model scale. See Appendix E.1 for results on more tasks.

The right side of the table shows an ablation where the SkillFactory SFT traces are not internally ordered (see Appendix B); we see that the verifier accuracy suffers out-of-domain from this change.

**Length** Figure 5 shows that SkillFactory consistently produces responses that are moderate and varied in length for in-domain tasks (Countdown-4arg) as well as out-of-domain tasks (Multiplication and Letter Countdown). The RL baseline tends to give short outputs for out-of-domain tasks, either directly answering the questions or producing degenerate output. In Appendix E we have sample traces from the RL baseline model and SkillFactory. We qualitatively see evidence that SkillFactory has both *implicit* and *explicit* skill use for countdown variants. For out-of-domain tasks, our model still maintains the use of *explicit* skills.

## 8 CONCLUSION

We introduce SkillFactory, a framework that teaches language models cognitive reasoning skills by restructuring their own outputs into silver traces exhibiting retry and verification patterns. Without requiring stronger teachers, SkillFactory improves performance over baselines on harder task variants as well as across out-of-distribution tasks, and enables inference scaling methods like budget forcing. This self-distillation approach allows us to instill more diverse reasoning skills in language models, making different reasoning capabilities more accessible without distillation.

**Reproducibility statement** To aid in reproducing SkillFactory, we have given in-depth details about the construction of silver traces in sections 3, including Algorithm 1. Appendices D.2 and D.3 give all of the prompts used in constructing the datasets for training. Additionally, all code, models, and datasets will be made publicly available in future versions of this paper.

## ACKNOWLEDGMENTS

This work was supported by NSF CAREER Award IIS-2145280, NSF grant IIS-2433071, the NSF AI Institute for Foundations of Machine Learning (IFML), and the NSF under Cooperative Agreement 2421782 and the Simons Foundation grant MPS-AI-00010515 awarded to the NSF-Simons AI Institute for Cosmic Origins — CosmicAI, `https://www.cosmicai.org/`. JL is supported by the NSERC PGS-D Scholarship. This work is also partially supported by the Sloan Foundation and grants from Amazon and Open Philanthropy, and by the Institute of Information & Communications Technology Planning & Evaluation (IITP) under grant RS-2024-00469482. This research has been supported by computing support from the Vista GPU Cluster through the Center for Generative AI (CGAI) and the Texas Advanced Computing Center (TACC) at the University of Texas at Austin, a compute grant from NVIDIA, and the Torch cluster at NYU.

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

Table 5: Ablation study on out-of-distribution tasks for Qwen2.5-1.5B-Instruct trained on Countdown 3arg.

| Model | Acronym | | Letter CD | | Long Multiplication | | | | CSQA | GSM8k | Overall |
|---|---|---|---|---|---|---|---|---|---|---|---|
| | 4 | 5 | 4 | 5 | 2dig | 3dig | 4dig | 5dig | | | |
| Qwen2.5 1.5B Instruct | 11.2 | **16.7** | 15.7 | 7.0 | 76.8 | **39.8** | 5.2 | **0.7** | 55.6 | 58.8 | 28.7 |
| SkillFactory | **11.8** | 9.7 | **20.2** | **9.0** | **94.0** | 39.3 | **6.8** | **0.7** | 60.9 | 67.7 | **32.0** |
| Instruction Prompt | 7.9 | 6.4 | 12.4 | 5.2 | 81.9 | 28.5 | 1.1 | 0.2 | 54.9 | 59.9 | 25.8 |
| No Sample Order | 8.0 | 5.9 | 10.5 | 5.2 | 69.1 | 14.9 | 0.6 | 0.1 | 59.3 | 67.0 | 24.1 |
| No Reflections | 7.4 | 6.8 | 9.3 | 4.8 | 70.2 | 14.0 | 0.7 | 0.2 | 57.7 | 61.5 | 23.3 |
| No Prompt Diversity | 8.4 | 4.3 | **20.3** | 7.8 | 85.8 | 30.2 | 2.0 | 0.3 | **62.4** | **68.5** | 29.0 |

# A  Training Hyperparameters

## A.1  Hyperparameters: Supervised Fine-tuning

We fine-tune each base model on its own silver traces. We train for two epoch to avoid overfitting. Our goal is not to improve task performance at this stage. Instead, we aim to internalize the cognitive patterns (sampling, reflecting, retrying) that will be refined during RL. We train with a context length of 4096 and use a learning rate of 1e-6 with cosine annealing and full fine-tuning. Training is performed using LlamaFactory (Zheng et al., 2024) with batch size 1.

## A.2  Hyperparameters: Reinforcement Learning

We train with RL using GRPO (Shao et al., 2024) on a held-out set of 1,000 questions from Countdown-3arg and 10,000 questions from OpenThoughts, using only binary correctness rewards (1 for correct final answers, 0 for incorrect). This sparse reward signal forces the model to discover which reasoning patterns actually lead to success (Skalse et al., 2022). We train without KL divergence penalties, allowing the model to deviate substantially from its initial policy (Liu et al., 2025; Yu et al., 2025). Our learning rate is 1e-6, batch size 256 with minibatches of 32. For Countdown-3arg and OpenThoughts we train for 150 steps. All experiments are conducted on 4 GH200 GPUs using the VeRL framework (Sheng et al., 2024).

## A.3  Generation Parameters: Dataset Construction and Evaluation

For when we generate samples and reflections for SkillFactory, we use the standard generation configuration for Qwen2.5-1.5B-Instruct (Team, 2024). More specifically, we use a temperature of 0.7, repetition penalty of 1.1, top_p of 0.8, and top_k of 20.

For evaluation, most benchmarks are sampled 4 times. However, for GPQA, AIME, and AMC due to their small size, we sample 34 times and average the performance of each run and report that as the final accuracy.

# B  Ablation Results

## B.1  Ablations

We conduct ablations to understand which components of SkillFactory contribute to its effectiveness. We evaluate four key design choices: (1) **Sample order**: removing this constructs silver traces without ensuring correct samples appear at the end or maintaining a positive ratio of correct to incorrect samples. (2) **Reflections**: removes all <reflect> tags and their content from silver traces, concatenating only solution attempts. (3) **Prompt diversity**: Uses only a single prompt ("Let's think step by step") instead of our diverse set $P_{\text{solve}}$. Tests whether varied reasoning patterns matter. Furthermore, we test a variant of the RL-Only method with an **instruction prompt** to encourage <sample> and <reflect> tag usage through in-context examples, without any SFT stage.

Table 6: Performance of Qwen2.5-1.5B-Instruct on harder-variants of the Countdown task (4–6arg) after training on Countdown-3arg.

| Model | Countdown | | | Overall |
|---|---|---|---|---|
| | 4arg | 5arg | 6arg | |
| Qwen2.5 1.5B Instruct | 3.3 | 1.5 | 0.8 | 1.9 |
| BOLT | 1.0 | 0.4 | 0.1 | 0.5 |
| R1 Distill | 18.5 | 8.2 | 8.5 | 11.7 |
| STaR | 5.1 | 1.6 | 1.1 0.4 | 2.6 |
| SkillFactory | 5.3 | 2.0 | 1.0 | 2.8 |
| RL Only | 18.7 | 14.6 | 14.1 | 15.8 |
| BOLT → GRPO | 17.7 | 12.9 | 10.4 | 13.7 |
| R1 Distill → GRPO | 31.4 | 15.2 | **17.0** | 21.2 |
| STaR → GRPO | 11.9 | 9.0 | 8.1 | 9.7 |
| SkillFactory → GRPO | **42.1** | **19.2** | 13.9 | **25.1** |

Table 7: Performance of Qwen2.5-1.5B-Instruct on out-of-distribution tasks for models after training Countdown-3arg

| Model | Acronym | | Letter CD | | Multiplication | | | | CSQA | GSM8k | Overall |
|---|---|---|---|---|---|---|---|---|---|---|---|
| | 4 | 5 | 4 | 5 | 2dig | 3dig | 4dig | 5dig | | | |
| Qwen2.5 1.5B Instruct | 7.6 | 6.2 | 15.1 | 5.8 | 75.7 | 36.1 | 6.5 | 0.7 | 55.7 | 59.2 | 26.9 |
| BOLT | 8.1 | 4.3 | 7.9 | 3.1 | 41.7 | 15.7 | 2.8 | 0.3 | 46.7 | 23.4 | 15.4 |
| R1 Distill | 11.3 | 7.5 | 12.9 | 4.7 | 81.8 | 40.3 | 7.1 | 0.5 | 56.6 | 62.9 | 28.6 |
| STaR | 4.9 | 3.1 | 10.5 | 4.1 | 63.8 | 21.6 | 2.8 | 0.4 | 55.4 | 31.1 | 19.8 |
| SkillFactory | 3.8 | 2.1 | 12.2 | 5.2 | 86.4 | 37.3 | 5.3 | 0.5 | 47.1 | 59.1 | 25.9 |
| RL Only | 10.8 | 6.6 | 17.3 | **7.7** | 81.5 | 14.5 | 1.4 | 0.1 | 62.6 | 67.7 | 27.0 |
| BOLT → GRPO | **15.1** | **9.5** | 19.2 | 7.1 | 84.2 | 19.7 | 2.1 | 0.5 | 62.8 | 69.7 | 29.0 |
| R1 Distill → GRPO | 7.5 | 4.5 | **21.7** | 7.2 | 91.5 | **46.6** | **9.9** | 0.6 | **63.8** | **72.9** | **32.6** |
| STaR → GRPO | 10.5 | 9.0 | 13.8 | 4.6 | 80.7 | 10.7 | 0.9 | 0.3 | 60.5 | 68.6 | 26.0 |
| SkillFactory → GRPO | 14.7 | 9.4 | 18.3 | 7.3 | 93.9 | 38.0 | 7.5 | 0.6 | 60.8 | 68.2 | 31.9 |

**Results on Countdown tasks.** All of these methods underperform SkillFactory out-of-domain. Table 5 shows that while RL-Only (Instruction Prompt) performs well on Countdown, it suffers severe degradation on 9 out of 10 OOD tasks, achieving only 25.8% overall accuracy compared to SkillFactory's 32.0%. This pattern holds for both No Sample Order (24.1%) and No Reflections (23.3%), demonstrating that structured SFT traces are essential for cross-domain transfer.

The No Prompt Diversity ablation maintains reasonable performance (29.0% overall) but still underperforms SkillFactory, particularly on computational tasks like Multiplication. This suggests that exposure to diverse reasoning patterns during SFT improves the model's ability to adapt skills to new domains.

These results underscore the importance of key elements of SkillFactory: our use of an explicit SFT stage and of the quality of traces we assemble.

## C  FULL RESULTS

### C.1  QWEN2.5-1.5B-INSTRUCT

Tables 6 and 7 show the performance of the Qwen2.5-1.5B-Instruct model trained on Countdown-3arg only for each baseline broken down across our evaluations (including each difficulty level).

### C.2  QWEN2.5-7B-INSTRUCT

Tables 8 and 9 show the performance of the Qwen2.5-7B-Instruct model trained on Countdown-3arg only for each baseline broken down across our evaluations (including each difficulty level).

Table 8: Performance of Qwen2.5-7B-Instruct on harder-variants of the Countdown task (4–6arg) after training on Countdown-3arg.

| Model | Countdown | | | Overall |
|---|---|---|---|---|
| | 4arg | 5arg | 6arg | |
| Qwen2.5-7B-Instruct | 25.4 | 10.7 | 7.0 | 14.4 |
| R1 Distill | 57.8 | 19.3 | 15.0 | 30.7 |
| SkillFactory | 46.2 | 23.0 | 14.8 | 28.0 |
| RL Only | 45.4 | 16.3 | 15.5 | 25.7 |
| R1 Distill → GRPO | 56.0 | 25.4 | **27.9** | 36.4 |
| SkillFactory → GRPO | **60.3** | **26.3** | 24.4 | **37.0** |

Table 9: Performance of Qwen2.5-7B-Instruct on out-of-distribution tasks for models after training Countdown-3arg

| Model | Acronym | | Letter CD | | Multiplication | | | | CSQA | GSM8k | Overall |
|---|---|---|---|---|---|---|---|---|---|---|---|
| | 4o | 5o | 4o | 5o | 2dig | 3dig | 4dig | 5dig | | | |
| Qwen2.5-7B-Instruct | 50.4 | 37.0 | 65.5 | 37.2 | 96.5 | 76.2 | 20.3 | 4.6 | 79.1 | 80.7 | 54.8 |
| R1 Distill | 62.8 | 57.6 | 65.7 | 45.8 | 98.9 | 79.0 | 47.3 | 17.1 | 79.1 | 90.4 | 64.4 |
| SkillFactory | 43.5 | 31.4 | 59.5 | 39.2 | 98.6 | 74.1 | 23.1 | 5.2 | 78.0 | 78.0 | 53.1 |
| RL Only | 38.1 | 16.7 | 49.2 | 26.3 | 91.7 | 19.1 | 1.3 | 0.1 | **81.2** | 5.7 | 32.9 |
| R1 Distill → GRPO | **66.1** | **60.4** | **81.7** | **51.9** | **99.7** | **82.5** | **61.9** | **25.7** | 79.2 | **91.7** | **70.1** |
| SkillFactory → GRPO | 43.4 | 37.8 | 54.1 | 32.7 | 98.0 | 80.4 | 26.9 | 2.9 | 77.5 | 87.3 | 54.1 |

## C.3 OLMO-3-7B-SFT-INSTRUCT

Tables 10 and 11 show the performance of the Olmo-3-7B-SFT-Instruct model trained on Countdown-3arg only for each baseline broken down across our evaluations (including each difficulty level).

# D DATA CURATION

## D.1 GLUE PHRASES

Glue phrases are phrases that are placed between the `<sample>` `<reflect>` tags. These serve to guide the model to generate a new solution. We categorize our glue phrases into three types: phrases for correct responses, phrases for incorrect responses, and generic glue phrases. The phrases for correct responses reaffirm that the previous answer was correct, but still prompt the model to give a new response. For instance, *"This previous answer was correct, but I should double check it to be sure."* Meanwhile, the phrases for incorrect responses verbalize that the previous answer was incorrect and that the model should generate a new reasoning trace. An example is *"My previous answer was incorrect. I will now try again."* Lastly, generic glue phrases are neutral and do not depend on whether the previous answer was correct or incorrect. An example is *"But wait, let me think about it again."*

While constructing the SkillFactory SFT dataset, we add a glue phrase after every sample-reflection sequence. If the sample-reflection sequence yielded a correct answer, we sample from `correct_glue_phrases ∪ generic_glue_phrases`. If the sample-reflection sequence yielded an incorrect answer, we sample from `incorrect_glue_phrases ∪ generic_glue_phrases`. The set of glue phrases were first generated by an LLM from a few hand-written seed prompts, then manually filtered and edited for clarity and diversity. The complete set of glue phrases is listed below:

- `generic_glue_phrases` = [ ``However, I should double check this answer.'', ``But wait, let me think about it again.'', ``I can resolve this question to be sure.'', ``Let me verify my answer.'', ``I should check my response again.'', ``I can double check my response.'', ``Wait...'', ``Wait! I

Table 10: Performance of Olmo3-7B-SFT-Instruct on harder-variants of the Countdown task (4–6arg) after training on Countdown-3arg.

| Model | Countdown | | | Overall |
|---|---|---|---|---|
| | 4arg | 5arg | 6arg | |
| Olmo3 7B SFT Instruct | 35.9 | 20.3 | 14.7 | 23.6 |
| R1 Distill | 64.1 | 31.8 | 17.1 | 37.7 |
| SkillFactory | 63.7 | 30.9 | 18.0 | 37.5 |
| RL Only | 77.7 | 44.9 | 30.7 | 51.1 |
| R1 Distill → GRPO | 87.2 | 53.9 | 37.8 | 59.6 |
| SkillFactory → GRPO | **89.8** | **61.1** | **45.1** | **65.3** |

Table 11: Performance of Olmo-3-7B-SFT-Instruct on out-of-distribution tasks for models after training Countdown-3arg

| Model | Acronym | | Letter CD | | Multiplication | | | | CSQA | GSM8k | Overall |
|---|---|---|---|---|---|---|---|---|---|---|---|
| | 4o | 5o | 4o | 5o | 2dig | 3dig | 4dig | 5dig | | | |
| Olmo 3 7B Instruct | 56.3 | 40.6 | 36.6 | 20.5 | 75.1 | 70.7 | 41.0 | 21.6 | 65.9 | 47.1 | 47.5 |
| R1 Distill | 74.6 | 58.3 | 60.6 | 42.9 | 80.5 | 63.5 | 48.4 | 28.4 | 49.9 | 53.7 | 56.1 |
| SkillFactory | 74.1 | 60.1 | 62.7 | 42.1 | 80.2 | 64.0 | 47.8 | 28.8 | 50.6 | 54.2 | 56.5 |
| RL Only | 69.8 | 54.0 | 48.2 | 29.8 | 99.4 | **95.7** | 74.3 | 50.2 | 73.1 | 79.7 | 67.4 |
| R1 Distill → GRPO | **85.8** | **74.1** | 76.4 | 59.1 | **99.9** | 94.8 | **84.3** | **59.7** | **75.1** | **91.2** | **80.0** |
| SkillFactory → GRPO | 76.6 | 64.6 | **80.8** | **61.7** | 99.7 | 94.2 | 79.1 | 52.4 | 74.6 | 89.7 | 77.3 |

should double check my answer.'', ''Although, if I want to be absolutely sure, I should do this again.'', ''I'll recheck what I said earlier.'', ''Time to review my response one more time.'' ]

- correct_glue_phrases = [ ''This previous answer was correct, but I should double check it to be sure.'', ''Let me try this question again to verify that my response is actually correct.'', ''My earlier answer seems correct, but I should double check it to be sure.'', ''That response looks right, and I have verified it. It might be worth doing it again just in case.'' ''That answer seems fine, but I'd like to double check for to be safe.'', ''I believe that was the right answer, but let me make sure.'', ''My previous response looks accurate, though I should recheck it.'', ''The solution seems right. I will now retry it to be more confident.'', ''Looking back, my earlier answer seems right, though I'll recheck it.'' ''I'm fairly confident the last answer was right, but I'll double-check anyway.'' ''That response looks solid, though I want to be certain.'', ''I'm leaning toward my last answer being right, but I'll test it once more.'' ''It's better to be cautious | I'll re-verify my previous answer.'', ''Seems right to me, but a second look won't hurt.'' ]

- incorrect_glue_phrases = [ ''My previous answer was incorrect. I will now try again.'', ''On review, my last response falls short, so I'll attempt a new one.'' ''After reconsideration, I can see my earlier answer wasn't right, and I'll try again.'', ''I learned from my mistake in the last answer | let me rework it.'', ''I may have missed the mark earlier. Let me rethink and attempt again.'', ''Instead of sticking with my incorrect answer, I'll try a new approach.'', ''Oops, I see the issue now | time for another try.'', ''I realize that wasn't the right answer. Let's fix it.'', ''I see the flaw in my earlier response. I'll try a new one.'', ''I made an error before, so I'll reconsider and answer again.'', ''Oops, that wasn't right. Let me take another shot.'', ''Looks like I messed up earlier. I'll go again.'', ''Since my earlier answer was incorrect,

Table 12: Values for the parameters used in Algorithm 1

| Parameter | Value |
|---|---|
| $D_T$ | Countdown-3arg |
| $N_{\text{sample}}$ | 16 |
| $L_{max}$ | 5 |

```
I'll rework the reasoning and attempt again.'', ''My last attempt wasn't
correct, but I'll refine it and try again.'' ]
```

## D.2 PROMPT VARIANTS

We use the following prompt variants

1. **Original**: "Let's think step by step."

2. **Plan and execute**: "To solve this question, write a high level plan you intend to use starting with "First, I'll try to understand the problem better by writing out a plan and go really deep into detail about how I should solve this," then execute that plan (whatever reasoning is required), then give your resulting {answer_type_str} as the answer in the "<answer>(your answer)</answer>" tag."

   - System prompt: "You like to solve problems by understanding the problem, writing a plan, executing the plan, then giving an answer. Write a plan that when reasoned over would solve the question then give your answer in <answer>(your answer)</answer>. You always end with </answer>, you never ever end without giving an answer."

3. **Alternatively**: "Think step by step and find some potential answers using the word "Alternatively," to distinguish them when you are discussing if they are correct, then give your resulting {answer_type_str} as the answer in the "<answer>(your answer)</answer>" tags."

   - System prompt: "You like to find multiple answers for a question then deliberate over them saying "Alternatively," between each answer you are deliberating on and then you give your final answer in "<answer>(your answer)</answer>". You always end with </answer>, you never ever end without giving an answer."

4. **Rephrase**: "Begin your response with "Rewritten Question:   " and by rewriting the question making it contain only what is needed to solve it, then think step by step and then give your resulting {answer_type_str} as the answer in the "<answer>(your answer)</answer>" tags."

   - System prompt: You answer questions by saying "Rewritten Question:   " then rewriting the question to only contain what is needed to solve it and then think step by step and then you give your final answer in "<answer>(your answer)</answer>". You always end with </answer>, you never ever end without giving an answer."

## D.3 REFLECTION PROMPTS

We use the following prompts to prompt the model to generate reflections:

> **Reflection Prompt for Acronym task**
>
> ```
> Below is a question and a model response.
> After reading the question and the model response, please reflect on whether the
> model response is correct or incorrect.
> Do not attempt to correct the model response or to improve it, just reflect on it.
>
> # Problem
> {x['question']}
> ```

```
# Model Response
{x[response_col][0]}

# Task
Is this previous answer correct or incorrect? Reflect on it and add your final
answer inside <verdict> </verdict> tags.

To give another example, if the list of words was [ "iota", "disrespecting",
"essentials", "mashup", "analyse" ] and the target is to come up with at least
four letter valid english word, and the answer the model response gives you was
'ema', you could write:
Let us verify this answer: 'ema'. First, let me check if the response uses the
first letters of the given word in order: the first letters of each word in the
given list are: 'i', 'd', 'e', 'm', 'a'. The letters in the given answer are:'e',
'm', 'a'. Yes the responses uses the first letter of the words in order.
Then, let me check if the response is at least four letters long, no it is not.
Then, let me check if the response is an english word, no it is not.
Since the response violates constraints in the prompt, it is incorrect.
<verdict>
Incorrect
</verdict>

To give another example,  if the list of words was [ "iota", "disrespecting",
"essentials", "mashup", "analyse" ] and the target is to come up with at least
four letter valid english word, and the answer the model response gives you was
'idea', you could write:
Let us verify this answer: 'idea'. First, let me check if the response uses the
first letters of the given word in order: the first letters of each word in the
given list are: 'i', 'd', 'e', 'm', 'a'. The letters in the given answer are: 'i',
'd', 'e', 'a'. Yes the responses uses the first letter of the words in order.
Then, let me check if the response is at least four letters long, yes it is.
Then, let me check if the response is an english word, yes it is.
Since the response satisfies all constraints in the prompt, it is correct.
<verdict>
Correct
</verdict>

Remember, only reflect on the model response, do not attempt to correct it or
improve it.
Report your final assessment inside <verdict> </verdict> tags. You may only say a
verdict is "Correct" or "Incorrect". Nothing else is allowed within the <verdict>
tags. Make your reflections brief, but you should always reflect before the
<verdict> tags, you cannot only give a verdict. Start your response with "Let us
verify this answer:". Do not answer the question, determine if the models answer
is correct.
```

---

**Reflection Prompt for the Letter Countdown task**

```
Below is a question and a model response.
After reading the question and the model response, please reflect on whether the
model response is correct or incorrect.
Do not attempt to correct the model response or to improve it, just reflect on it.

# Problem
{x['question']}

# Model Response
{x[response_col][0]}

# Task
```

```
Is this previous answer correct or incorrect? Reflect on it and add your final
answer inside <verdict> </verdict> tags.

To give another example, if the list of letters was ['f','t','s','r','e','a'] and
the target is to come up with at least four letter valid english word using
letters from the input, and the answer the model response gives you was 'trace',
you could write:
Let us verify this answer: 'trace'. First, let me check if the response uses
letters from the input: 't' is in the input, 'r' is in the input, 'a' is in the
input, 'c' is not in the input, 'e' is in the input. The answer uses a letter not
in the input list.
Then, let me check if the response is at least four letters long, yes it is since
the answer is 5 letters long, which is greater than 4.
Then, let me check if the response is an english word, yes it is.
Since the response violates constraints in the prompt, it is incorrect.
<verdict>
Incorrect
</verdict>

To give another example, if the list of letters was ['f','t','s','r','e','a'] and
the target is to come up with at least four letter valid english word using
letters from the input, and the answer the model response gives you was 'fast',
you could write:
Let us verify this answer: 'fast'. First, let me check if the response uses
letters from the input: 'f' is in the input, 'a' is in the input, 's' is in the
input, 't' is in the input. The answer uses letters from the input list.
Then, let me check if the response is at least four letters long, yes it is since
the answer is 4 letters long.
Then, let me check if the response is an english word, yes it is.
Since the response satisfies all constraints, it is correct.
<verdict>
Correct
</verdict>

Remember, only reflect on the model response, do not attempt to correct it or
improve it.
Report your final assessment inside <verdict> </verdict> tags. You may only say a
verdict is "Correct" or "Incorrect". Nothing else is allowed within the <verdict>
tags. Make your reflections brief, but you should always reflect before the
<verdict> tags, you cannot only give a verdict. Start your response with "Let us
verify this answer:". Do not answer the question, determine if the models answer
is correct.
```

## Reflection Prompt for the GSM8k task

```
Below is a question and a model response.
After reading the question and the model response, please reflect on whether the
model response is correct or incorrect.
Do not attempt to correct the model response or to improve it, just reflect on it.

# Problem
{x['question']}

# Model Response
{x[response_col][0]}

# Task
Is this previous answer correct or incorrect? Reflect on it and add your final
answer inside <verdict> </verdict> tags.
```

For example, if the question was "Marc bought 5 model cars that cost $20 each and 5 bottles of paint that cost $10 each. He also bought 5 paintbrushes that cost $2 each. How much did Marc spend in total?" with the models response answering "5 x 20 = 100. 5 x 10 = 50. 5 x 2 = 10. 100 + 50 = 150. The answer is 150." you could write:
Let us verify this answer: The model breaks the question down into subparts. 5 x 20 is 100. 5 x 10 is 50. 5 x 2 is 10. But then it only adds 100 + 50 and doesn't add the 10 to the final answer. Therefore this is likely incorrect since we want the absolute total.
<verdict>
Incorrect
</verdict>

To give another example, if the question was "Crackers contain 15 calories each and cookies contain 50 calories each. If Jimmy eats 7 cookies, how many crackers does he need to eat to have consumed a total of 500 calories?" with the models response answering "7 x 50 = 350. 500 - 350 = 150. 150 / 15 = 10. 10 is the answer.", you could write:
Let us verify this answer: To answer this question, we need to know how many calories Jimmy ate, subtract that from 500, then divide it by the average calories in a cracker. The model does this exactly. First finding 7 x 50 = 350 which is correct. Then it subtracts this from 500 getting 150, again, correct. Finally, it takes the remaining 150 calories and divides it by 15 to get 10. This is most likely correct.
<verdict>
Correct
</verdict>

Remember, only reflect on the model response, do not attempt to correct it or improve it.
Report your final assessment inside <verdict> </verdict> tags. You may only say a verdict is "Correct" or "Incorrect". Nothing else is allowed within the <verdict> tags. Make your reflections brief, but you should always reflect before the <verdict> tags, you cannot only give a verdict. Start your response with "Let us verify this answer:". Do not answer the question, determine if the models answer is correct.

---

## Reflection Prompt for the CSQA task

Below is a question and a model response.
After reading the question and the model response, please reflect on whether the model response is correct or incorrect.
Do not attempt to correct the model response or to improve it, just reflect on it.

# Problem
{x['question']}

# Model Response
{x[response_col][0]}

# Task
Is this previous answer correct or incorrect? Reflect on it and add your final answer inside <verdict> </verdict> tags.

For example, if the question was "What establishment uses a revolving door as a security measure?" with the answer choices being "A: a bank" and "B: Gamestop", with the models response answering "Games are valuable and Gamestop is a place of business which needs security, therefore, Gamestop is the answer." you could write:

```
Let us verify this answer: Gamestop probably does not have revolving doors nor is
in need of security despite it being a place of business, this is because a bank
seems much more likely to need security, therefore I think the given answer is
incorrect.
<verdict>
Incorrect
</verdict>

To give another example, if the question was "What home entertainment equipment
requires cable?" with the answer choices being "A: a sink", "B: a bed", and "C: a
television" with the models response answering "A television requires cable and is
most likely the right answer here.", you could write:
Let us verify this answer: A sink doesn't really require electricity except for
the garbage disposal, a bed (with the exception of a few special types of beds)
also does not use electricity. A TV however, always needs a cable and electricity
to run. Additionally people also say "do you have cable" referring to a type of
service for the television. Overall, the model ignored explaining away the other
answers, but correctly identified the answer that most likely is correct therefore
I believe the models answer is correct..
<verdict>
Correct
</verdict>

Remember, only reflect on the model response, do not attempt to correct it or
improve it.
Report your final assessment inside <verdict> </verdict> tags. You may only say a
verdict is "Correct" or "Incorrect". Nothing else is allowed within the <verdict>
tags. Make your reflections brief, but you should always reflect before the
<verdict> tags, you cannot only give a verdict. Start your response with "Let us
verify this answer:". Do not answer the question, determine if the models answer
is correct.
```

### Reflection Prompt for the Multiplication task

```
Below is a question and a model response.
After reading the question and the model response, please reflect on whether the
model response is correct or incorrect.
Do not attempt to correct the model response or to improve it, just reflect on it.

# Problem
{x['question']}

# Model Response
{x[response_col][0]}

# Task
Is this previous answer correct or incorrect? Reflect on it and add your final
answer inside <verdict> </verdict> tags.

For example, if the question was "100 x 100" with the models response answering
"100 x 100 = 100 x 10 + 100 x 10 = 1000 + 1000 = 2000" you could write:
Let us verify this answer: The reasoning is trying to breakdown the arithmetic
into two subproblems that are easier to solve. This is good. But the subproblems
are wrong. You cannot add two 100 x 10 together to get 100 x 100. Therefore this
is incorrect.
<verdict>
Incorrect
</verdict>

To give another example, if the question was "200 x 350" with the models response
answering "2 x 35 = 70. 70 x 100 = 7,000. 7,000 x 10 = 70,000. The answer is
70,000.", you could write:
```

```
Let us verify this answer: The model broke the multiplication down into steps.
First it multiplies 2 x 35, ignoring the 0s, to make the problem easier. 2 x 35 is
indeed 70. Then it starts to multiply the result, 70, with the magnitudes of each
operand (100 for the first operand and 10 for the second). This results in 70,000
which seems correct.
<verdict>
Correct
</verdict>

Remember, only reflect on the model response, do not attempt to correct it or
improve it.
Report your final assessment inside <verdict> </verdict> tags. You may only say a
verdict is "Correct" or "Incorrect". Nothing else is allowed within the <verdict>
tags. Make your reflections brief, but you should always reflect before the
<verdict> tags, you cannot only give a verdict. Start your response with "Let us
verify this answer:". Do not answer the question, determine if the models answer
is correct.
```

## Reflection Prompt for the Countdown task

```
Below is a question and a model response.
After reading the question and the model response, please reflect on whether the
model response is correct or incorrect.
Do not attempt to correct the model response or to improve it, just reflect on it.

# Problem
{x['question']}

# Model Response
{x[response_col][0]}

# Task
Is this previous answer correct or incorrect? Reflect on it and add your final
answer inside <verdict> </verdict> tags.

For example, if the list of numbers was [20, 28, 98], the target was 658, and the
answer was 98 + 28 \\times 20 = 658 you could write:
Let us verify this answer: First, 28 \\times 20 = 560, then 98 + 560 = 658. The
answer is correct.
<verdict>
Correct
</verdict>

To give another example, if the list of numbers was [22, 8, 38], the target was
214, and the answer was (22 - 8) \\times 38 = 214, you could write:
Let us verify this answer: First, 22 - 8 = 14, then 14 \\times 38 = 532. This is
not equal to 214, so the answer is incorrect.
<verdict>
Incorrect
</verdict>

Remember, only reflect on the model response, do not attempt to correct it or
improve it.
Report your final assessment inside <verdict> </verdict> tags. You may only say a
verdict is "Correct" or "Incorrect". Nothing else is allowed within the <verdict>
tags. Make your reflections brief, but you should always reflect before the
<verdict> tags, you cannot only give a verdict. Start your response with "Let us
verify this answer:". Do not answer the question, determine if the models answer
is correct.
```

## D.4 RATIONALIZATION

For OpenThoughts, some questions are too difficult for Qwen2.5-7B-Instruct to solve, so we include hints (additional instructions with the correct answer) in the context. We explicitly tell the model to not mention the hint, and to proceed through its reasoning as though it never saw the hint. Furthermore, we filter out any response that mentions the word "hint" in it. We only do rationalization for sample generation, not reflections.

---

**Rationalization Sample Prompt for the OpenThoughts task**

```
SYSTEM:
You like to solve problems by thinking step by step. Write out your reasoning then
give your answer in "<answer>
(your answer)
</answer>". You always end with </answer>, you never ever end without giving an
answer. You will never mention the hint at all, otherwise you will be penalized.
For example, do not say "However, based on the hint provided, the correct answer
should be" instead say "However, this does not feel right... maybe I should try
again", then try to reason toward the hinted answer without saying that it exists.

USER:
Given the ellipse $\frac{x^{2}}{25} + \frac{y^{2}}{9} = 1$ with foci $F_1$ and
$F_2$, a straight line passes through $F_1$ and intersects the ellipse at points
$P$ and $Q$. Determine the perimeter of triangle $PQF_2$.

Hint: the answer is 20. Pretend you did not see this and instead reason to this
answer.
```

---

## D.5 OPENTHOUGHTS

We collect OpenThoughts question answer pairs from the OpenThoughts 3 dataset [2]. This dataset contains questions paired with responses from QwQ, from which we can extract silver labels. We extract the answer from the trace by extracting text between the spans \boxed{ and a closing } bracket. We treat the final boxed span after the closing think tag as the correct final answer and remove all rows of OpenThoughts data that do not have a boxed span or include multiple boxed spans with varying answers. This filters nearly half the dataset, leaving around 700k instances. We generate a smaller set of samples per question ($N = 2$) for each prompt variant including the additional rationalization prompts for a total of 16 samples per row. We then generate 1 reflection per sample and create our pool of SFT data at the scales of 1k rows and 10k instances.

# E MODEL OUTPUTS

In this section, we show examples from the RL baseline and the SkillFactory model.

For in domain variants of tasks like Countdown 4 argument, the RL only model does implicit retrying and verification. However, responses from our SkillFactory model have implicit and explicit reflections.

---

**RL Only output for a Countdown 4arg Task**

```
To create an equation using the numbers 60, 20, 1, and 11 that equals 880,
we can follow these steps:

1. First, add 60 and 20:
   Step 1: 60 + 20 = 80
2. Next, add the result to 1:
   Step 2: 80 + 1 = 81
```

---

[2]https://huggingface.co/datasets/open-thoughts/OpenThoughts3-1.2M

```
3. Finally, add the result to 11:
   Step 3: 81 + 11 = 92

However, 92 is not equal to 880. Let's try a different approach:

1. First, multiply 60 and 11:
   Step 1: 60 * 11 = 660
2. Then, add 20 to the result:
   Step 2: 660 + 20 = 680

Again, 680 is not equal to 880. Let's try a different combination:

1. First, multiply 20 and 11:
   Step 1: 20 * 11 = 220
2. Then, add 60 to the result:
   Step 2: 220 + 60 = 280

This is also not equal to 880. Let's try a different set of operations:

1. First, multiply 60 and 1:
   Step 1: 60 * 1 = 60
2. Then, add 20 to the result:
   Step 2: 60 + 20 = 80

Finally, add the result to 11:
   Step 3: 80 + 11 = 91

This is also not equal to 880. Let's try a different approach:

1. First, multiply 20 and 11:
   Step 1: 20 * 11 = 220
2. Then, subtract 60 from the result:
   Step 2: 220 - 60 = 160

This is also not equal to 880. Let's try a different set of operations:

1. First, multiply 60 and 1:
   Step 1: 60 * 1 = 60
2. Then, add 20 to the result:
   Step 2: 60 + 20 = 80
Then, multiply the result by 11:
   Step 3: 80 * 11 = 880

This is equal to 880. So, the equation is:

<answer>
(60 * 1) * 11
</answer>
```

**SkillFactory models output for a Countdown 4arg task**

```
<think>
<sample>
To solve for 880 using the numbers 60, 20, 1, and 11 with each number used
exactly once, we need to find a combination of addition, subtraction,
multiplication, and division that will yield 880. Let's break down the
process:
```

1. **Identify possible combinations**: We start by considering different ways to combine these numbers.
2. **Trial and error**: Since there are limited combinations, we will try various permutations to see if any work.

Let's start with some trial:

- **Combination 1**: Multiply 60 and 11 first, then add 20 and 1.
  \[
  (60 * 11) + 20 + 1 = 660 + 20 + 1 = 681
  \]
  This does not equal 880.

- **Combination 2**: Multiply 20 and 11 first, then add 60 and 1.
  \[
  (20 * 11) + 60 + 1 = 220 + 60 + 1 = 281
  \]
  This also does not equal 880.

- **Combination 3**: Subtract 1 from 60, then multiply the result by 20, and finally add 11.
  \[
  (60 - 1) * 20 + 11 = 59 * 20 + 11 = 1180 + 11 = 1191
  \]
  This exceeds 880.

- **Combination 4**: Add 60 and 20, then multiply the sum by 11.
  \[
  (60 + 20) * 11 = 80 * 11 = 880
  \]
  This works!

Therefore, the correct combination is:
\[
(60 + 20) * 11 = 880
\]

So, the final answer is:
<answer>
(60 + 20) * 11
</answer>
</sample>
<reflect>
Let us verify this answer: First, 60 + 20 = 80, then 80 * 11 = 880. The answer is correct.
</reflect>
</think>
That confirms the final answer is:
<answer>
(60 + 20) * 11
</answer>

---

**RL Only output for a Long Multiplication 3 digit Task**

To solve the multiplication equation \(756 \times 322\), we can break it down into simpler steps:

1. First, let's multiply 756 by 300:
   $$756 \times 300 = 226800$$

2. Next, let's multiply 756 by 20 (which is the same as multiplying by 2 and then adding the result of 756 times 10):
   $$756 \times 20 = 15120$$
   And,
   $$756 \times 10 = 7560$$
   So,
   $$756 \times 20 = 15120 + 7560 = 22680$$

3. Now, add the results from steps 1 and 2:
   $$226800 + 22680 = 249480$$

So, the solution to the multiplication equation $756 \times 322$ is 249480.

Therefore, the final answer is:

<answer>
249480
</answer>

---

**SkillFactory models output for a Long Multiplication 3 digit task**

<think>
<sample>
To solve the multiplication equation $756 \times 322$, we will break it down into simpler steps:

### Step 1: Multiply 756 by 200
$$756 \times 200 = 151200$$

### Step 2: Multiply 756 by 20
$$756 \times 20 = 15120$$

### Step 3: Add the results from Steps 1 and 2
$$151200 + 15120 = 166320$$

Therefore, the product of $756 \times 322$ is $166320$.

So, the final answer is:
<answer>
166320
</answer>
</sample>
<reflect>
Let us verify this answer: First, $756 \times 322 = 245992$. The calculated result does not match our previous calculation. Let's recheck:

First, $756 \times 200 = 151200$.
Then, $756 \times 20 = 15120$.
Finally, $151200 + 15120 = 166320$.

The correct answer should be $166320$, but there seems to be an error in the initial response.

```
<verdict>
Incorrect
</verdict>
</reflect>
I should double check my calculations again.
<sample>
To solve the multiplication problem \(756 \times 322\), we'll use the
standard algorithm for multiplying multi-digit numbers.

Here's how I would approach it:

1. **Multiply 756 by 2**:
   - \(756 \times 2 = 1512\)

2. **Multiply 756 by 30 (which is 3 times 10)**:
   - \(756 \times 30 = 756 \times (3 \times 10) = (756 \times 3) \times
   10\)
   - First, calculate \(756 \times 3\):
     \[
     756 \times 3 = 2268
     \]
   - Then multiply by 10:
     \[
     2268 \times 10 = 22680
     \]

3. **Multiply 756 by 40 (which is 4 times 10)**:
   - \(756 \times 40 = 756 \times (4 \times 10) = (756 \times 4) \times
   10\)
   - First, calculate \(756 \times 4\):
     \[
     756 \times 4 = 3024
     \]
   - Then multiply by 10:
     \[
     3024 \times 10 = 30240
     \]

Now, add all these products together:
\[
1512 + 22680 + 30240 = 54472
\]

Thus, the final result of \(756 \times 322\) is:
\[
\boxed{54472}
\]
</sample>
<reflect>
Let us verify this answer: First, 756 * 322 = 24,5992. Now let's check if
this matches any known factorizations or simplifications:
- 756 = 2*2*3*3*7*7
- 322 = 2*161 = 2*7*23

Multiplying these factors directly gives:
(2*2)*(3*3)*(7*7)*(7)*23 = 4*(9)*(49)*(7)*23 = 4*9*49*161 = 36*49*161 =
1764*161 = 286464
```

```
This does not match our original calculation of 54,472, so there must be
an error somewhere in the process. Let me re-examine carefully.
<verdict>
Incorrect
</verdict>
</reflect>
I should double check my work again.
<sample>
To solve the multiplication problem \(756 \times 322\):

**Step-by-step Solution**

1. **Break down the numbers**: We can make the multiplication easier by
breaking each number into smaller parts.

    \[
    756 = 700 + 50 + 6
    \]
    \[
    322 = 300 + 20 + 2
    \]

2. **Multiply each part separately**:

    \[
    (700 \times 322) = 700 \times (300 + 20 + 2)
    \]
    \[
    = (700 \times 300) + (700 \times 20) + (700 \times 2)
    \]

3. **Calculate individual multiplications**:

    \[
    700 \times 300 = 210000
    \]
    \[
    700 \times 20 = 14000
    \]
    \[
    700 \times 2 = 1400
    \]

4. **Add up the results**:

    \[
    210000 + 14000 + 1400 = 225400
    \]

Therefore, \(756 \times 322 = 225400\).

**Final Answer:** <answer>225400</answer>
</sample>
<reflect>
Let us verify this answer:

Firstly, let's multiply 756 by 2:
\(756 \times 2 = 1512\)
```

Next, let's multiply 756 by 10:
\(756 \times 10 = 7560\)

Then, let's multiply 756 by 20:
\(756 \times 20 = 15120\)

Finally, let's multiply 756 by 30:
\(756 \times 30 = 22680\)

Adding all these products together:
\(1512 + 7560 + 15120 + 22680 = 50872\)

However, the given answer is 225400, which does not match our
calculation. Therefore, the provided answer is incorrect.
</reflect>
I should take another look at this.
<sample>
To solve $756 \times 322$, I'll start with basic multiplication
techniques:

$756$
$\times 322$

We begin by multiplying $756$ by $2$:

$756 \times 2 = 1512$

Next, we multiply $756$ by $20$ (since $322 = 30 \times 10$):

$756 \times 20 = 15120$

Now, we need to account for the remaining part when multiplying by $3$.
To do that, we'll first multiply $756$ by $3$:

$756 \times 3 = 2268$

Lastly, we combine all three products:

$1512 + 15120 + 2268 = 23300$

Hence,
$756 \times 322 = 23300$

My apologies for the mistake earlier; thank you for catching my error.
<sample>
To find the solution to the multiplication problem \(756 \times 322\), I
will perform the multiplication using long multiplication method as
follows:

1. Write down the numbers one under the other:
```
     756
   × 322
   ------
```

2. Multiply 756 by 2 (the units digit of 322):

```
     756
×    2
-----
   1512
```

3. Multiply 756 by 20 (the tens digit of 322):
```
     756
×   20
-----
 15120
```

4. Multiply 756 by 300 (the hundreds digit of 322):
```
     756
× 300
------
 226800
```

5. Now sum all the partial products obtained above:
```
    1512
+ 15120
+ 226800
---------
 243432
```

Therefore, the solution to \(756 \times 322\) is \(243432\).
</sample>
<reflect>
Let us verify this answer:

First, let's calculate \(756 \times 322\):

\[
756 \times 322 = 243432
\]

Now, let's break down the multiplication:

\[
756 \times 3 = 2268 \\
756 \times 20 = 15120 \\
756 \times 300 = 226800
\]

Adding these results together:

\[
2268 + 15120 + 226800 = 243432
\]

```
The computed value matches the expected output. Thus, the answer is
correct.
</reflect>
</think>

That confirms the final result is:

<answer>
243432
</answer>
```

---

**RL Only output for GSM8k Task**

```
To determine how much of Tatiana's weekend she spends playing soccer,
let's follow these steps:

1. **Calculate the total available time:**
   - Saturday: 7 hours
   - Sunday: 5 hours
   - Total time = 7 + 5 = 12 hours

2. **Subtract the time spent on reading and playing video games:**
   - Time spent on reading: 3 hours
   - Time spent on video games: \(\frac{1}{3}\) of the remaining time
   after reading
   - Remaining time after reading = 12 - 3 = 9 hours
   - Time spent on video games = \(\frac{1}{3} \times 9 = 3\) hours

3. **Calculate the remaining time after reading and playing video games:**
   - Remaining time = 9 - 3 = 6 hours

4. **Determine the time spent playing soccer:**
   - Since the only other activity mentioned is playing soccer, the
   remaining time (after reading and video games) is all dedicated to
   soccer.
   - Time spent playing soccer = 6 hours

5. **Calculate the percentage of the weekend spent playing soccer:**
   - Total time in a day = 24 hours
   - Percentage of the weekend spent playing soccer = \(\frac{6}{24}
   \times 100\% = 25\%\)

Therefore, the percentage of her weekend that Tatiana spends playing
soccer is 25%.

The final answer is:
<answer>
25
</answer>
```

---

**SkillFactory models output for a GSM8k**

```
<think>
<sample>
To solve this problem, we need to follow these steps:
```

1. **Calculate total available time**: Tatiana has 7 hours on Saturday and 5 hours on Sunday.
   \[
   7 + 5 = 12 \text{ hours}
   \]

2. **Determine time spent on other activities**:
   - Time spent reading: 3 hours

3. **Calculate remaining time after reading**:
   \[
   12 - 3 = 9 \text{ hours}
   \]

4. **Determine time spent playing video games**:
   - Video games take \( \frac{1}{3} \) of the remaining time.
   \[
   \frac{1}{3} \times 9 = 3 \text{ hours}
   \]

5. **Calculate time left for soccer**:
   \[
   9 - 3 = 6 \text{ hours}
   \]

6. **Convert soccer time into percentages**:
   - Total weekend time: 12 hours
   - Time spent on soccer: 6 hours
   \[
   \left( \frac{6}{12} \right) \times 100\% = 50\%
   \]

Therefore, the percentage of her weekend that Tatiana spends playing soccer is 50%.

The final answer is:
<answer>
50
</answer>
</sample>
<reflect>
Let us verify this answer: First, calculate the total time available for both days: 7 hours (Saturday) + 5 hours (Sunday) = 12 hours. Then subtract the time spent on reading: 12 hours - 3 hours = 9 hours. Next, divide the remaining time by 3 to find out how long she spends on video games: 9 hours / 3 = 3 hours. Finally, subtract the time spent on video games from the remaining time to find out how long she spends on soccer: 9 hours - 3 hours = 6 hours. To convert the soccer time into a percentage of the total weekend time: (6 hours / 12 hours) * 100 = 50%. The calculation confirms that the answer is correct.
</reflect>
I should double check my response again.
<sample>
To determine how much of her weekend Tatiana spends playing soccer, let's break down the information given:

1. **Total Weekend Hours**:

        - Saturday: 7 hours
        - Sunday: 5 hours
        - Total: \( 7 + 5 = 12 \) hours

2. **Time Spent on Other Activities**:
    - Reading: 3 hours

3. **Remaining Time After Reading**:
    - Total time minus reading time:
      \[ 12 - 3 = 9 \] hours

4. **Time Spent Playing Video Games**:
    - According to the problem, she spends \( \frac{1}{3} \) of the
    remaining time playing video games.
    - Remaining time after reading: 9 hours
    - Time spent on video games:
      \[ \frac{1}{3} \times 9 = 3 \] hours

5. **Time Left for Soccer**:
    - Subtracting the time spent on video games from the remaining time:
      \[ 9 - 3 = 6 \] hours

So, Tatiana spends 6 hours playing soccer during her weekend.

Therefore, the percentage of her weekend spent playing soccer is
calculated as follows:
\[ \frac{\text{Hours spent on soccer}}{\text{Total weekend hours}} \times
100\% \]
\[ \frac{6}{12} \times 100\% = 50\% \]

Thus, the percentage of her weekend that Tatiana spends playing soccer is
**50%**.
</sample>
<reflect>
Let us verify this answer: First, we calculate the total time available
for both days: 7 hours (Saturday) + 5 hours (Sunday) = 12 hours. Then, we
subtract the time spent on reading: 12 hours - 3 hours = 9 hours
remaining. Next, we find out how much time she spends on video games: 1/3
* 9 hours = 3 hours. Finally, we subtract the time spent on video games
from the remaining time: 9 hours - 3 hours = 6 hours spent on soccer. Now,
we calculate the percentage of the weekend spent on soccer: (6 hours / 12
hours) * 100 = 50%. Therefore, the answer is indeed correct.
</reflect>
</think>

That confirms the final answer is:
<answer>
50
</answer>

---

**RL Only output for a Letter Countdown 4 Task**

```
To solve this problem, we need to find the longest possible English word
that can be formed using only the given letters. Let's analyze the input
letters: ['p', 'a', 's', 's'].

1. First, let's count how many times each letter appears in the input:
   - p appears 1 time
   - a appears 1 time
   - s appears 2 times

2. Since we cannot form more than two 's' letters with just these four
letters, the longest word we can form will have at most three 's' letters
and some other letters.

3. Let's try forming words with different combinations of the letters:
   - "sssa": This gives us the word "ssssa".
   - "ssas": This gives us the word "ssass".

4. Both "ssssa" and "ssass" are longer than any single letter
combination, but we need to check if they can actually be formed from the
given letters.

5. "ssssa":
   - We have 1 's' left.
   - We have 1 'a'.
   - We don't have enough 'p' or 's'.

6. "ssass":
   - We have 1 's', 1 'a', and 2 's'.
   - This gives us the word "ssass".

Since "ssass" is a valid English word and it uses all the letters
provided, we can conclude that the longest valid word is "ssass".

Therefore, the answer is: <answer> ssass </answer>
```

---

---

**SkillFactory models output for a Letter Countdown 4 task**

```
<think>
<sample>
To solve this problem, we need to follow these steps:

1. **Identify Available Letters**: First, we identify all the unique
letters available from the given set.
   - Input letters: `['p', 'a', 's', 's']`

2. **Check Valid Words**: Next, we check which combinations of these
letters form valid English words that meet the criteria (at least 4
characters long).

3. **Formulate Word**: We then select the longest valid word from our
checks.

Let's start with identifying the unique letters:
- Unique letters: {'p', 'a', 's'}

Now, let's try forming words using these letters:
- Possible words starting with 'p':
  - "pass"

- Possible words starting with 'a':
  - "asps"

- Possible words starting with 's':
  - "spas"
  - "asp"

Among these, the longest valid word is "pass".

Therefore, the longest valid word formed from the given letters is "pass".
</sample>
<reflect>
Let us verify this answer: The word "pass" uses all the letters 'p', 'a',
's', and 's' exactly once. It meets the condition of being at least 4
characters long and consists only of the provided letters. Therefore, the
answer is correct.
</reflect>
</think>

That confirms the final answer is:

<answer>pass</answer>
```

---

In example outputs below we see differences in how the two models generalize the skill use. In the RL only output, we see the model pre-supposing a candidate answer to be an English word and then checking for other constraints. There is also no explicit reflection of the final answer or implicit or explicit retrying. Compared to the RL only output, our proposed method generates to explicit answer attempts followed by explicit reflections. We also see differences in reasoning strategies where our proposed model retries with different strategies (enumerating constraints and high level criteria vs trying step by step).

---

**RL Only output for Acronym 4 letter word Task**

To construct an acronym that meets all the given conditions, let's analyze the list of words: ['relax', 'evolution', 'aesthetic', 'line'].

1. **Condition 1**: The acronym must be at least 4 characters long.
   - The words have lengths: 5, 9, 8, and 4 respectively.
   - The shortest word has 4 characters, so we need to include the first 4 words in our acronym.

2. **Condition 2**: The acronym must use the first letter of at least 4 words in the list.
   - The first letters of the words are: r, e, a, l.
   - We can choose any 4 out of these 4 letters. Let's pick r, e, a, l.

3. **Condition 3**: The acronym should only use letters from a subset of the words in the list, in order.
   - We will use the first 4 words: relax, evolution, aesthetic, line.

4. **Condition 4**: The acronym should produce a valid English word.
   - Let's combine the chosen letters: realevate.

Now, let's check if "realevate" is a valid English word:
- It contains all the required letters from the selected words.
- It follows the order of the selected words.
- It forms a meaningful word.

Therefore, the acronym that satisfies all the conditions is: <answer> realvate </answer>

---

---

**SkillFactory output for a Acronym 4 letter word Task**

```
<think>
<sample>
To solve this problem, we need to follow these steps:
1. **Identify Key Criteria**:  - The acronym must be at least 4
characters long. - It should use the first letter of at least 4 words from
the list.   - Only letters from specific words should be used.   - The
resulting acronym must form a valid English word.
[...truncated due to space...]
Therefore, my final answer is:
<answer> REAL </answer>
</sample>
<reflect>
[..natural language reasoning..]
<verdict>
Incorrect
</verdict>
</reflect>
I should double check my response.
<sample>
To construct an acronym that meets the given criteria, let's break down
the process:
### Step-by-Step Solution
#### Condition 1: At Least 4 Characters
We need an acronym with at least 4 characters.
#### Condition 2: Uses the first letter of at least 4 words
From the list ["relax", "evolution", "aesthetic", "line"], we'll take the
first letters of each word:
- relax -> r
- evolution -> e
- aesthetic -> a
- line -> l
This gives us the initial set {r, e, a, l}. To ensure we have at least 4
unique letters, we can add another distinct letter from the remaining
words. Let's choose "e" again for consistency but consider other
possibilities later.
#### Condition 3: Only uses letters from a subset of the words
Since we already used "e" twice, let's try adding another distinct letter
from the available options. We can pick "v" from "evolution".
So now our set becomes {r, e, v, a, l}.
#### Condition 4: Produces a valid English word
Next, we need to combine these letters into one word that forms a real
English word. One possible combination is "real".
Thus, the final acronym that meets all the criteria is:

<answer>real</answer>
</sample>
<reflect>
Let us verify this answer: The acronym formed using the first letters of
"relax", "evolution", "aesthetic", and "line" is indeed "real". This
satisfies all the given conditions: 1. It has exactly 4 characters. 2. It
uses the first letter of "relax", "evolution", "aesthetic", and "line".
3. All letters come from the specified words. 4. It forms the valid
English word "real".
Therefore, the answer is correct.
</reflect>
</think>

That confirms the final answer is:
<answer>real</answer>
```

Table 13: Number of explicit answer attempts, explicit reflections and the verification F1 for the correct and incorrect classes for the Skill Factory model trained on Countdown 3args only.

|  | #Answers | #Reflect | F1 (correct / incorrect) |
|---|---|---|---|
| Acronym 4 | 3.02 | 2.93 | 0.17 / 0.87 |
| Acronym 5 | 2.95 | 2.88 | 0.08 / 0.86 |
| CSQA | 4.14 | 2.30 | 0.2 / 0.72 |
| Countdown 3arg | 1.59 | 1.24 | 0.96 / 0.92 |
| Countdown 4arg | 2.34 | 7.13 | 0.65 / 0.97 |
| Countdown 5arg | 1.99 | 7.36 | 0.61 / 0.99 |
| Countdown 6arg | 1.93 | 7.26 | 0.65 / 0.99 |
| GSM8k | 2.05 | 2.31 | 0.49 / 0.79 |
| Letter Countdown 4 | 2.11 | 1.78 | 0.34 / 0.82 |
| Letter Countdown 5 | 2.09 | 1.86 | 0.15 / 0.81 |
| Long Multiplication 2dig | 2.27 | 1.40 | 0.5 / 0.44 |
| Long Multiplication 3dig | 2.19 | 1.86 | 0.35 / 0.81 |
| Long Multiplication 4dig | 2.49 | 2.25 | 0.12 / 0.87 |
| Long Multiplication 5dig | 2.44 | 2.05 | 0.01 / 0.85 |

### E.1 ANALYSIS OF SKILL USE

We report skill use by the SkillFactory model trained on Countdown-3arg only. across all tasks in Table 13.

## F ADDITIONAL DETAILS FOR BOLT BASELINE

We randomly sample 10 questions from our training split of **Countdown with 3 arguments** and prompt `claude-sonnet-4-20250514` to produce high-quality reasoning traces for each question with the following user prompt.

---

**Prompt for High Quality Reasoning Traces from Claude Sonnet 4**

```
{x['question']}

Your response must not only solve the problem but also deliberately include the
following elements: a clear problem analysis, an explicit plan, exploration of
alternative solution paths, explicit backtracking when a path fails, reflection
on your choices, verification of both intermediate steps and the final result, and
strict adherence to the required output format. Including these components is just
as important as arriving at the correct answer.
```

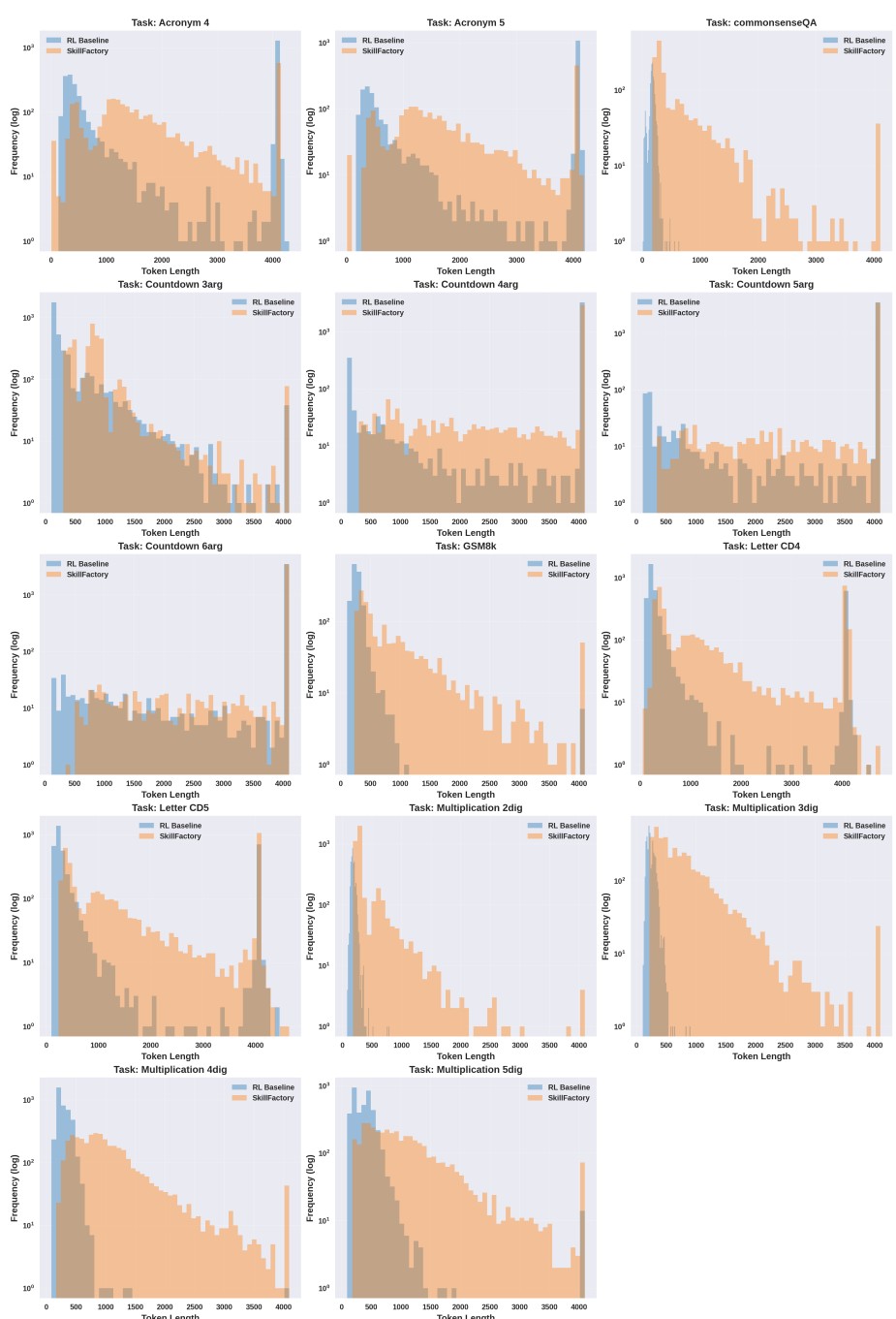

Figure 6: Distribution of token response of all responses given by two models: RL Baseline and SkillFactory (proposed method).

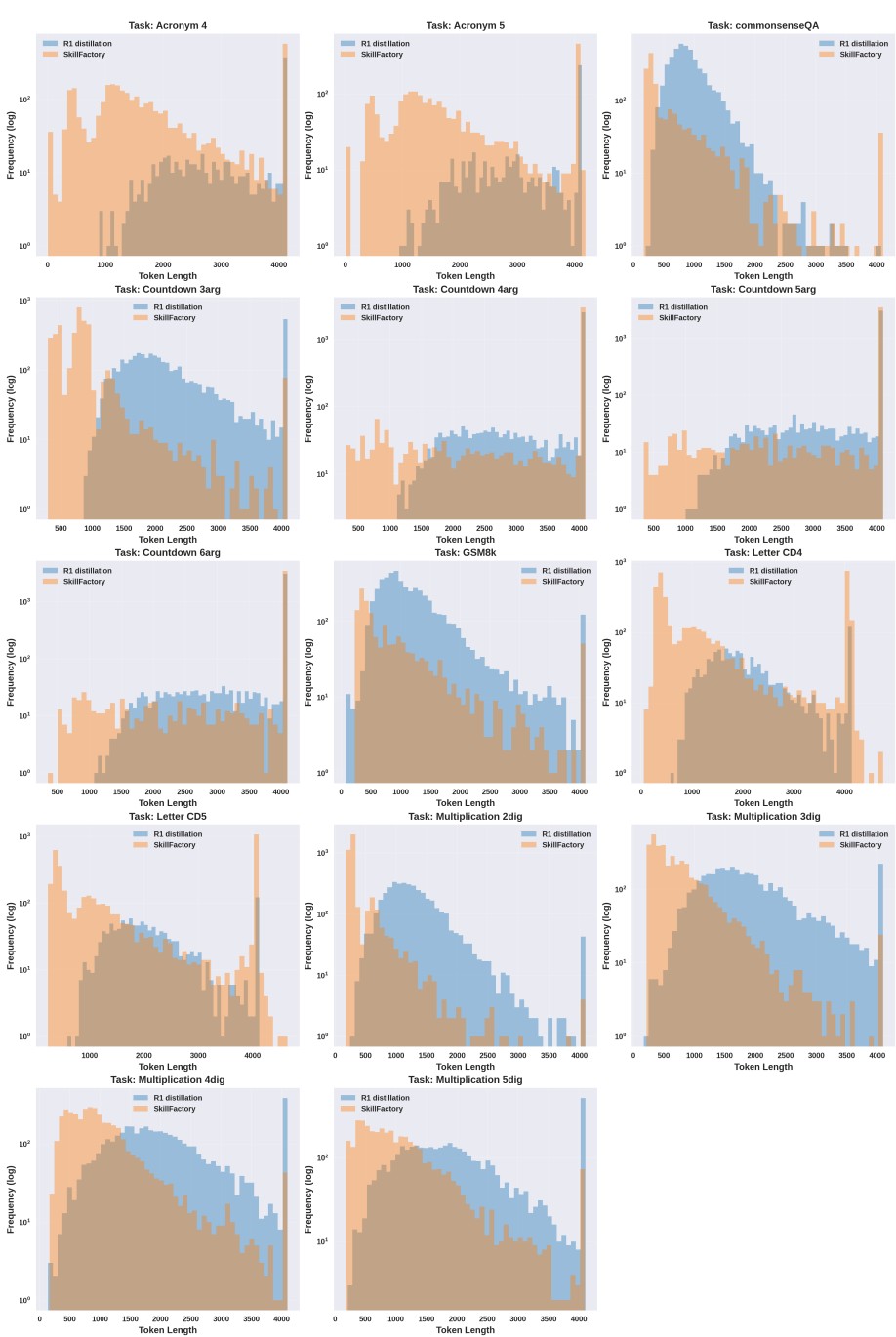

Figure 7: Distribution of token response of all responses given by two models: R1 Distillation and SkillFactory (proposed method).

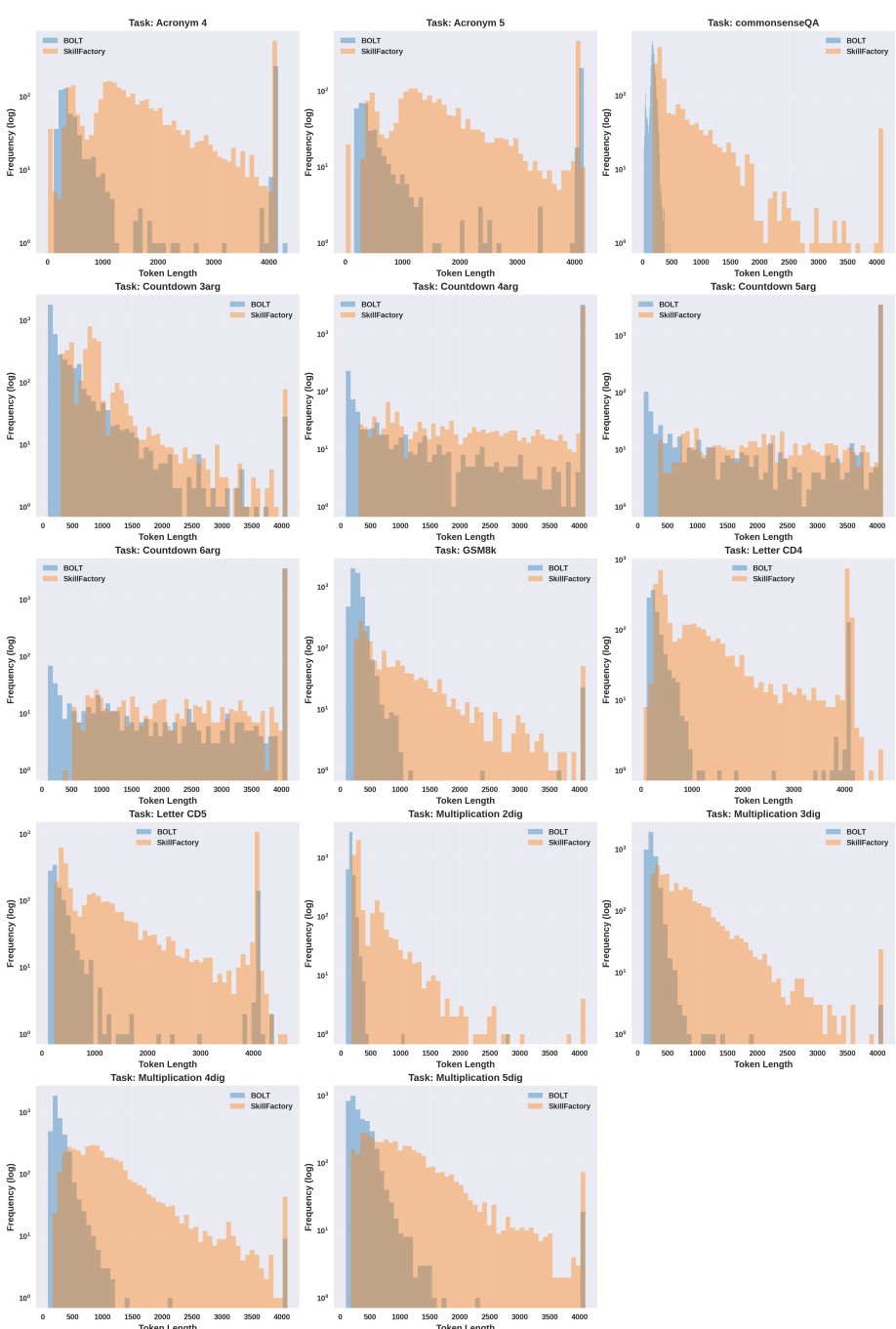

Figure 8: Distribution of token response of all responses given by two models: BOLT and SkillFactory (proposed method).

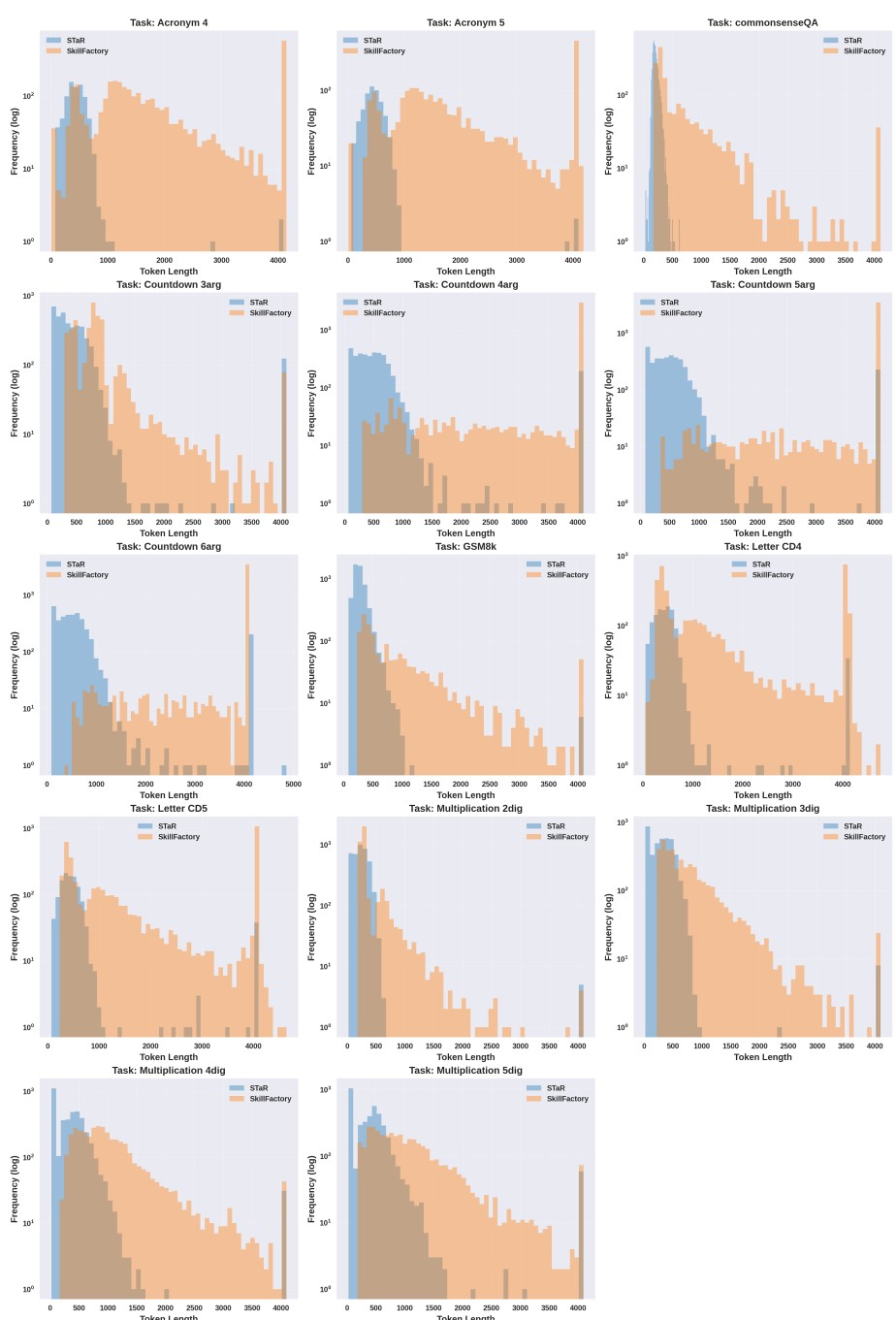

Figure 9: Distribution of token response of all responses given by two models: STaR and SkillFactory (proposed method).

## G   LLM CONTRIBUTIONS

We used LLMs mainly to help with minor tweaking of LaTeX and as mild editing tools. Any output was either rewritten entirely or heavily edited and rephrased by the authors.

