# OpenReview forum: "SkillFactory: Self-Distillation for Learning Cognitive Behaviors"
_ICLR.cc/2026/Conference — ICLR 2026 Poster_

### Official Review · Reviewer_3nnB · 2025-10-31

**Soundness:** 2
**Presentation:** 3
**Contribution:** 2
**Rating:** 4
**Confidence:** 4

**Summary:**

This paper proposes SkillFactory, a self-distillation method that trains small models to learn reasoning skills like reflection and retrying by rearranging their own outputs into structured “silver” traces. After supervised fine-tuning and reinforcement learning, the model shows improved reasoning and generalization on Countdown and OOD tasks without relying on larger teacher models.

**Strengths:**

1. The paper is well-written and clearly organized, making it easy to follow.

2. The data generation process is detailed, combining multiple prompting strategies to self-elicit diverse cognitive skills in small models as well as to ensure diversity.

3. The experiments are comprehensive, covering both in-domain and out-of-distribution evaluations, with detailed length analysis and ablation studies that strengthen the empirical support.

**Weaknesses:**

1. While the idea to equip models with cognitive skills is clear, the current formulation focuses on a small, pre-defined set of tagged skills (e.g., retry and reflection). As these skills are incorporated through explicit templates and tags, it limits the generalization to other cognitive behaviors beyond these predefined patterns.

2. SkillFactory involves both SFT and RL stages, whereas most baselines rely on only one of these training efforts (except STaR). The training effort is not consistent across these methods which leads to unfair comparison.

3. Although SkillFactory’s main strength emerges after the RL stage, its SFT performance remains notably below the R1-distilled baseline (Table 1). While R1-distilled traces are expected to yield stronger results even after RL, the paper does not report such comparisons (both in-domain and OOD), making it difficult to assess how close SkillFactory’s trajectories come to matching teacher-distilled SFT data.

**Questions:**

1. Countdown-3arg seems to be a relatively easy dataset, with performance saturating quickly after training. It seems that SkillFactory assumes the base model is at least sufficiently capable to generate several correct answers during sampling. Would the method still be effective if the base model struggles to produce correct outputs on more challenging datasets?

2. The experiments are conducted using a single model (Qwen2.5-1.5B-Instruct) trained on one dataset (Countdown). Have the authors trained on harder datasets or larger models to assess its generalizability?

3. How is the number of reflection/retry steps determined when composing a SkillFactory trajectory?

---

> ### Author Response · Authors · 2025-12-03
> **Response to Reviewer 3nnB**
>
> Thank you for your comments and feedback! We’d like to address some of the questions and concerns below:
>
> > the current formulation focuses on a small, pre-defined set of tagged skills […] it limits the generalization to other cognitive behaviors beyond these predefined patterns.
>
> SkillFactory encourages high level cognitive skills such as retry and verification, but it does not limit the skills that the base model may surface through standard RL within those skills. For example, we see search and backtracking being used especially for the Countdown domain in both RL only and SkillFactory despite not having an explicit tag for either of these skills; these occur organically within the <answer> tag. Secondly users can always define more skills with more tags following Section 3. We believe that these two points actually help the generalization to other cognitive behaviors (both latently as well as give the ability to create new skills with new custom tags). In addition, we do not see any evidence or reason to assume SkillFactory cannot generalize to more skills.
>
> >  The training effort is not consistent across these methods which leads to unfair comparison.
>
> We include post-SFT and post-RL results for all baselines in the new version of the paper to help make this clearer. We note that in our original draft, only R1 distillation was not trained using RL (all others were). We now include R1 Distill with GRPO in the new version of the paper.
>
> > making it difficult to assess how close SkillFactory’s trajectories come to matching teacher-distilled SFT data.
>
> We added a new experiment for R1 Distillation with GRPO. We found that SkillFactory can outperform both the R1 Distillation and R1 Distillation with GRPO baselines for easy-to-hard generalization (held-out countdown tasks). We also find that in the OpenThoughts experiment, SkillFactory remains competitive to using expert generated traces from QwQ. We believe this is very encouraging, as it frees us from the use of much larger teacher models.
>
> > Would the method still be effective if the base model struggles to produce correct outputs on more challenging datasets?
>
> Great question! As it stands, we need to be able to sample both correct and incorrect generations from the model.  Without this, the model has either saturated the task or cannot solve the task at all, and in both cases SkillFactory cannot be used as is. However, this can be overcome using “rationalization”, which we used in our OpenThoughts experiment (see the general response). Rationalization produces reasoning post-hoc given the correct answer. Because SkillFactory uses prompts from the base model, one could include the correct answer in your prompt and have the model “reason to the answer” instead of solving the problem itself allowing SkillFactory to be used on more challenging datasets than the base model itself can solve.
>
> > The experiments are conducted using a single model (Qwen2.5-1.5B-Instruct) trained on one dataset (Countdown).
>
> Please see the general response; we introduce 2 additional models and a new training setting that encompasses multiple domains such as challenging math, code, and science.
>
> > How is the number of reflection/retry steps determined when composing a SkillFactory trajectory?
>
> We do not have a controlled method for selecting this other than an indirect constraint on how many samples (of the 64) were correct and incorrect (if the model primarily gets a question correct, there may be a limited number of incorrects or retries). Ablating trace construction on heuristics such as question difficulty would be an exciting direction, but we have left this as future work.

---

### Official Review · Reviewer_pNtV · 2025-11-01

**Soundness:** 2
**Presentation:** 4
**Contribution:** 2
**Rating:** 4
**Confidence:** 3

**Summary:**

This paper presents a framework for teaching language models cognitive reasoning skills without requiring stronger teacher models. The key insight is that models can learn these skills from rearranged "silver" traces constructed from their own outputs. The method involves three stages: (1) sampling diverse solutions and reflections from a base model, (2) rearranging these into structured traces with explicit skill markers using tags, and (3) supervised fine-tuning followed by reinforcement learning. SkillFactory achieves substantial improvements on harder variants and better generalization to out-of-distribution tasks.

**Strengths:**

- The idea of creating structured "silver" training data by rearranging a model's own outputs is impactful.
- The paper proposes a sound training pipeline that shows compelling generalization evidence to other tasks. Experiments are done in extensive settings. (RL Only, STaR, BOLT, R1 Distillation).

**Weaknesses:**

- The entire study uses only Qwen2.5-1.5B-Instruct. Larger models (7B+) may already exhibit these skills naturally. In addition, larger models might have better performance on some baselines’ settings (such as reinforcement learning). The generalizability of conclusions in this paper beyond 1.5B parameters is highly uncertain.
- Training exclusively on Countdown 3-arg (where solutions are easy to verify but hard to find) is an ideal scenario for reflection/verification skills. The paper lacks discussion of how SkillFactory's effectiveness changes for tasks where verification itself is difficult, subjective, or computationally expensive (e.g., creative writing, legal arguments).

**Questions:**

- Have you tested with at least one larger model (e.g., Qwen2.5-7)? This is critical to understand whether SkillFactory remains beneficial at practical scales where models may already exhibit some skills naturally.
- Have you considered an experiment where SkillFactory is trained on a mixture of tasks (e.g., Countdown + GSM8K + Multiplication) rather than just Countdown? This would clarify whether the approach is domain-general or specific to search-like problems, which could change my assessment of the method's practical utility.

---

> ### Author Response · Authors · 2025-12-03
> **Response to Reviewer pNtV**
>
> Thank you for your feedback and we are excited that you agree that SkillFactory’s data construction process is impactful. We hope to address your questions and concerns below:
>
> > The entire study uses only Qwen2.5-1.5B-Instruct. Larger models (7B+) may already exhibit these skills naturally.
>
> > Have you tested with at least one larger model (e.g., Qwen2.5-7)?
>
> > Have you considered an experiment where SkillFactory is trained on a mixture of tasks
>
> Please see our general response; we introduce two additional models (including Qwen2.5-7B-Instruct) and a more realistic data setting consisting of challenging math, science, and coding questions.
>
> > verification itself is difficult, subjective, or computationally expensive (e.g., creative writing, legal arguments).
>
> As mentioned in our comment above, we do expand into more domains. However, moving into traditionally non-verifiable domains we are leaving for future work, primarily because we believe SkillFactory is one of many steps to get non-verifiable domains to work well for reasoning models.  That being said, we are extremely excited by this direction and glad you see SkillFactory’s potential in these areas as we do!

---

### Official Review · Reviewer_KXRf · 2025-11-01

**Soundness:** 2
**Presentation:** 3
**Contribution:** 2
**Rating:** 4
**Confidence:** 4

**Summary:**

This work presents a self-distillation framework that constructs silver SFT traces by sampling multiple responses from a base model (without instruction following fine-tuning), forming a long context-style SFT data with reflections. The proposed method conducts experiments that use these silver SFT data as a warm-up of the RL stage. Experiments show the model trained with warm-up data can have better OOD performance on harder variants of toy and real tasks, such as Countdown, 3-digits multiplication.

**Strengths:**

1. The writing is clear and well-organized.
2. Related work is comprehensive, and the authors carefully position SkillFactory relative to RL-only, distillation from stronger models, and self-distillation methods.

**Weaknesses:**

1. The title "SkillFactory" suggests a broad capability to learn diverse cognitive skills, but the implemented pipeline focuses on retrying and reflection. As L171 states, the proposed method has three steps: sampling diverse solutions, generating reflections, and assembling structured traces. It does not convincingly involve a wide variety of skills beyond long CoT with explicit verification and retry.
2. Several prior works adopt a similar idea of sampling multiple attempts and assembling them into longer, structured traces with reflections, such as injecting "wait" to force the model to think more or combine multiple traces. The author discussed these methods in the related works, and makes the claim that "SkillFactory is similar to these methods, but focuses on generating data entirely from the base model and highlights that structure is key for the generalization of consistent skill use." Firstly, I appreciate that the author made a clear statement of the relation against prior works. However, this motivation may be insufficient for a top-tier conference paper. Why does a similar idea purely rely on the base model to make a new method?
3. The main results take the Qwen2.5-1.5B-Instruct and focus on toy tasks, like Countdown, digit multiplication, and Letter. This limits the strength of the claims about general-purpose reasoning skill acquisition. Results on widely accepted long-CoT benchmarks, such as AIME/AMC/GPQA-Diamond with 7B size models, would substantially improve the paper.

**Questions:**

see weakness

---

> ### Author Response · Authors · 2025-12-03
> **Response to Reviewer KXRf**
>
> Thank you for detailed feedback! We have tried to address your concerns and questions as listed below:
>
> > It does not convincingly involve a wide variety of skills beyond long CoT with explicit verification and retry.
>
> We believe the overall method of SkillFactory (SFTing on data showing a particular type of skill, which is then refined in RL) is more general than what we show. It is not straightforward to adapt this to a wide variety of skills, which is why we don’t give results on that in this paper. We believe this is possible in principle, though, and emphasize our excitement about the underlying, general method.
>
>
> > Why does a similar idea purely rely on the base model to make a new method?
>
> A key motivation for our research is to illustrate a method that can work when distillation is *not* possible, e.g., when used on top of the largest model in its class (i.e. there was nothing to distill R1 from other than R1 itself when it was released).  Additionally, our results show that distillation from larger models (such as R1) into smaller models like Qwen2.5 1.5B Instruct may create a better reasoner post-SFT but it does not create a better reasoner post RL than our method does. So there is a practical reason to restrict ourselves to this setting and also results that confirm the viability of this approach compared to distillation. We think this deviates substantially from current practice in a way that’s worth highlighting.
>
> > The main results take the Qwen2.5-1.5B-Instruct and focus on toy tasks
>
> Please see the general response; we introduce two additional models and a realistic datasetting for both training and evaluation.

---

### Official Review · Reviewer_VLtu · 2025-11-02

**Soundness:** 3
**Presentation:** 3
**Contribution:** 3
**Rating:** 6
**Confidence:** 5

**Summary:**

The paper proposes a synthetic data generation technique for self-distilling reasoning behaviors into language models. The authors show that their method, SkillFactory, helps LLMs learn these behaviors by training on the game of countdown. they also show transfer to other domains like Common Sense QA, GSM8k etc.

**Strengths:**

- The paper is clearly written and the method is straightforward.
- The generalization performance of the model trained on 3 dig countdown is impressive.
- The method matches or surpasses distillation from stronger models.
- The paper also has good ablations and baselines.

**Weaknesses:**

- The biggest weakness of the paper is that all experiments are with the qwen2.5-1.5B model. Adding a model from another family and a different model size would help show the generalizability of the method.
- I have some questions about circularity: How do you reconcile necessity of the method if behaviors are already present in the pretraining data? “skills surface less consistently.” If the model can generate correct solutions and reflections (required for silver traces), why can't RL alone elicit these behaviors?
- If the reasoning behaviors aren’t present in the pretraining data, how does the model produce those?
- The paper mentions silver traces "may contain errors" but provides no analysis of error rates, types, or impact on learning.

**Questions:**

See weaknesses.

---

> ### Author Response · Authors · 2025-12-03
> **Response to Reviewer VLtu**
>
> Thank you for your comments and feedback! We’d like to address some of the questions and concerns below:
>
> > If the model can generate correct solutions and reflections (required for silver traces), why can't RL alone elicit these behaviors?
>
> The only requirement for SkillFactory trace generation is that a model be able to answer questions correctly. Models do not have to organically demonstrate chaining solutions and reflections together. This is the benefit of SkillFactory: the framework allows us to teach models this skill from scratch.
>
> This has two main ramifications. First, this enables learning of skills in domains where they aren’t naturally exhibited. Second, our paper shows that the learned skills are more generalizable than skills learned via RL alone (see Fig 5 about trace length).
>
>
> > The biggest weakness of the paper is that all experiments are with the qwen2.5-1.5B model
>
> See our general response comment, which introduces results from 2 additional models.
>
> > If the reasoning behaviors aren’t present in the pretraining data, how does the model produce those?
>
> Our trace construction procedure is detailed in Section 3. Note that the approach there does not assume the ability of the base model to demonstrate the skills, only that the base model can generate correct answers. The trace generation procedure bootstraps from these “basic” traces to produce SkillFactory traces that impart the skill via SFT and RL.
>
> > The paper mentions silver traces "may contain errors" but provides no analysis of error rates, types, or impact on learning.
>
> The notion of “error” is somewhat hard to define here. For instance, a model might produce the correct answer, double-check it, produce an incorrect answer, then double-check that, produce the correct answer, and stop. We’re not sure whether to call this an error. It is to some extent, but it is also an expected result of test time compute; we expect more computation to lead to correct answers, but not always in a clean way.
>
> As a result, we have largely sidestepped trying to define exactly what an error is in a trace. We will clarify this point in any future version.

---

### Author Response · Authors · 2025-12-03
**New Experiments & Results**

# Summary
Thanks to all the reviewers for their thoughtful comments and feedback! We responded to individual comments below, but comment here on our new draft and new experiments we ran. These experiments address a common thread across reviewers, namely to include more base models (all reviewers) and additional or more challenging settings (reviewers KXRf, 3nnB, and pNtV).
We note for context that R1 distillation is not an apples-to-apples comparison with our method. SkillFactory uses a self-distillation approach (training only from traces sampled from an initial model), whereas all approaches labeled with R1 assume access to a highly-capable reasoning model for distillation.

# New Experiments



## SkillFactory works on multiple base models

We show additional evaluations for Qwen2.5-7B-Instruct and Olmo-3-7B-Instruct-SFT. These experiments still focus on training on Countdown-3arg and evaluating the effectiveness of each model on harder held-out variants of the task as well as out of domain tasks.
For each data condition, we show the performance of all methods post-SFT and post-SFT+RL.

### Results

**Overall accuracy on Countdown (CD) and Out-of-Distribution (OOD) tasks:**

| Method | Qwen-1.5B CD | Qwen-1.5B OOD | Qwen-7B CD | Qwen-7B OOD | Olmo-7B CD | Olmo-7B OOD |
|:---|:---:|:---:|:---:|:---:|:---:|:---:|
| Base | 1.9 | 26.9 | 14.4 | 54.8 | 23.6 | 47.5 |
| RL Only | 14.4 | 27.0 | 25.7 | 32.9 | 51.1 | 67.4 |
| R1 | 11.7 | 28.6 | 30.7 | 64.4 | 37.7 | 56.1 |
| R1 → GRPO | 21.2 | **32.6** | 36.4 | **70.1** | 59.6 | **80.0** |
| SkillFactory | 2.8 | 25.9 | 28.0 | 53.1 | 37.5 | 56.5 |
| SkillFactory → GRPO | **25.1** | 31.9 | **37.0** | 54.1 | **65.3** | 77.3 |

Our results show that SkillFactory consistently leads to better easy-to-hard generalization on the held-out countdown tasks (4–6 arg) for all models, even surpassing R1 distillation with GRPO.  While R1 distillation with GRPO often performs better out-of-domain, SkillFactory remains competitive and is better than RL only. **These findings show that SkillFactory can improve performance across larger models and model classes.**


## SkillFactory works in more challenging data conditions

We further experimented using Qwen2.5-7B-Instruct trained on a subset of the OpenThoughts dataset, altering it to be a question answer pair dataset so it could be used by SkillFactory. We compare our baselines, including the original OpenThoughts distillation data (from QwQ), to SkillFactory as well as RL only. We train on subsets of the OpenThoughts dataset varying the size of the SFT data by 1k to 10k and evaluate on GPQA, AIME25, AMC, and Math500.

### Results

**Performance on Hard Math benchmarks:**

| Model | GPQA | AIME 25 | AMC | Math500 | Overall |
|:---|:---:|:---:|:---:|:---:|:---:|
| RL Only | 53.8 ± 1.6 | 5.4 ± 1.2 | 33.5 ± 0.8 | 59.1 ± 0.8 | 38.0 |
| QwQ with 1k rows | 48.5 ± 1.7 | 10.6 ± 1.4 | 19.9 ± 0.8 | 55.2 ± 0.9 | 33.5 |
| QwQ with 10k rows | **59.5** ± 1.5 | **15.3** ± 1.0 | 36.5 ± 0.9 | 58.6 ± 0.8 | **42.5** |
| SkillFactory with 1k rows | 56.7 ± 1.5 | 9.7 ± 1.4 | **37.5** ± 0.8 | **64.6** ± 0.7 | 42.1 |
| SkillFactory with 10k rows | 57.9 ± 1.5 | 7.3 ± 1.2 | 35.2 ± 0.7 | 61.9 ± 0.7 | 40.6 |

SkillFactory enhances reasoning capabilities on challenging math and science datasets. We find that at the 10k scale, SkillFactory reaches an overall score of 40.6%, closely approaching QwQ distillation (42.5%). At the 1k scale, **SkillFactory performs competitively across tasks and surpasses QwQ distillation on AMC (37.5%) and Math500 (64.6%)**, two benchmarks not explicitly targeted in the original OpenThoughts curation. In contrast, QwQ distillation exhibits degradation on Math500 relative to the base model even at 10k.

We note that SkillFactory’s performance slightly decreases from 1k to 10k examples (42.1\% to 40.6\%). We believe additional SFT does not help SkillFactory because the core skills are already learned early, unlike in distillation, where models learn new strategies and knowledge from the teacher.


# New Draft
We have uploaded a new version of the paper with the following changes:

1. We include experiments with additional models varying in size and class, specifically Qwen2.5-7B-Instruct and Olmo-3-7B-SFT.
2. We include results in the OpenThoughts setting , evaluating on datasets such as GPQA, AIME 2025, AMC, and Math500 to help expand our findings of SkillFactory beyond training on Countdown-3arg only.
3. We added an R1 Distill with GRPO baseline and find that SkillFactory outperforms this baseline across the models we evaluate on the held-out Countdown tasks, showing better easy-to-hard generalization while remaining competitive on out-of-domain tasks.

---

### Meta-Review · Area_Chair_Dudh · 2026-01-06

**Summary:**

In the original reviews, the reviewers' primary concerns centered around the limited scope of the experimental validation, notably the reliance on a single small model (Qwen2.5-1.5B-Instruct) and a narrow task (Countdown-3arg), which raised doubts about the generalizability and practical impact of the SkillFactory method. Additional critiques included questions about the method’s novelty compared to existing self-distillation or reflection-based techniques, the lack of error analysis for the generated "silver" traces, potential circularity in skill acquisition, and whether the approach would remain effective for larger models that might naturally exhibit such reasoning skills or for tasks where verification is inherently difficult.

The authors' rebuttal has effectively addressed most of the concerns, so I recommend Accept.

**Reviewer Concerns:**

The authors' rebuttal effectively addressed several key concerns by introducing new experiments with larger and more diverse base models (Qwen2.5-7B-Instruct and Olmo-7B) and expanding evaluation to more challenging, multi-domain benchmarks (e.g., GPQA, AIME25, AMC, Math500). They also clarified the distinction between SkillFactory and teacher-distillation methods, emphasizing its utility when no stronger teacher model is available.

**Reviewer Scores:**

After checking the authors' rebuttal, I feel that nearly all the reviewers would likely increase their scores, given the authors' response and new experimental results.

---

### Decision · Program_Chairs · 2026-01-26

Accept (Poster)